# PSMBENCH: A Benchmark and Dataset for Evaluating LLMs Extraction of Protocol State Machines from RFC Specifications

**Zilin Shen**
Purdue University
610 Purdue Mall, West Lafayette, IN 47907
shen624@purdue.edu

**Xinyu Luo**
Purdue University
610 Purdue Mall, West Lafayette, IN 47907
luo466@purdue.edu

**Imtiaz Karim**
University of Texas at Dallas
800 W Campbell Rd, Richardson, TX 75080
imtiaz.karim@utdallas.edu

**Elisa Bertino**
Purdue University
610 Purdue Mall, West Lafayette, IN 47907
bertino@purdue.edu

## Abstract

Accurately extracting protocol-state machines (PSMs) from the long, densely written Request-for-Comments (RFC) standards that govern Internet-scale communication remains a bottleneck for automated security analysis and protocol testing. In this paper, we introduce RFC2PSM , the first large-scale dataset that pairs 1,580 pages of cleaned RFC text with 108 manually validated states and 297 transitions covering 14 widely deployed protocols spanning the data-link, transport, session, and application layers. Built on this corpus, we propose PSMBENCH, a benchmark that (i) feeds chunked RFC to an LLM, (ii) prompts the model to emit a machine-readable PSM, and (iii) scores the output with structure-aware, semantic fuzzy-matching metrics that reward partially correct graphs.

A comprehensive baseline study of nine state-of-the-art open and commercial LLMs reveals a persistent state–transition gap: models identify many individual states (up to 0.82 F1) but struggle to assemble coherent transition graphs ($\leq 0.38$ F1), highlighting challenges in long-context reasoning, alias resolution, and action/event disambiguation. We release the dataset, evaluation code, and all model outputs as open-sourced[1], providing a fully reproducible starting point for future work on reasoning over technical prose and generating executable graph structures. RFC2PSM  and PSMBENCH aim to catalyze cross-disciplinary progress toward LLMs that can interpret and verify the protocols that keep the Internet safe.

## 1   Introduction

The current generation of large language models (LLMs) shows impressive ability to convert natural-language instructions into structured outputs, including tables, JSON records, and executable code. The conversion of technical specifications into *Protocol State Machines* (PSMs) remains an unsolved real-world challenge because PSMs serve as graph-structured abstractions for *security fuzzing* [Pham et al., 2020, De Ruiter and Poll, 2015], *formal verification* [Cremers et al., 2017, Beurdouche et al., 2017], and *implementation testing* of network protocols [Chen et al., 2023, Park et al., 2022, Pacheco

---

[1]Our dataset and benchmark are at RFC_PSM_Benchmark repository , promoting transparency and reproducibility in the community.

39th Conference on Neural Information Processing Systems (NeurIPS 2025) Track on Datasets and Benchmarks.

et al., 2022a], such as TCP [Eddy, Wesley (Ed.), 2022] and FTP [Postel and Reynolds, 1985]. Creating PSMs from Request for Comments (RFC) documents through manual methods requires months of time and specialized domain knowledge, which blocks automated security analysis [Graham and Johnson, 2014].

Existing approaches, such as mGPTFuzz [Ma et al., 2024], are based on manually selecting a protocol standard's relevant sections to generate the corresponding state machine. In addition, they are often protocol-specific - e.g. mGPTFuzz is specific to the an Internet of Things protocol [Connectivity Standards Alliance, 2022]. Another approach - PROSPER [Sharma and Yegneswaran, 2023], shows that prompted GPT-3.5 helps to identify states and transitions in limited RFCs. However, there is a lack of a standardized benchmark for measuring extraction quality across diverse specifications, and a gap in the automated measurement of graph-level transition system accuracy at scale.

Consequently, there is an urgency for a *standardised, diverse* testbed to evaluate the ability of the state-of-the-art LLM models OpenAI [2023], DeepSeek-AI et al. [2025], DeepMind [2024] in reasoning over lengthy technical prose and generate executable graph structures [Pacheco et al., 2022a]. An open-sourced comprehensive benchmark and dataset with many protocols is also critical to evaluating the ability of LLMs to extract PSMs from a wide range of standard protocol documents.

To meet this need, we construct a comprehensive dataset, RFC2PSM , a curated corpus of 14 RFCs spanning application, session, transport, and link-layer protocols, paired with manually validated ground-truth PSMs. Building on this corpus, we also introduce PSMBENCH, the first benchmark that (i) feeds chunked RFC text to an LLM, (ii) asks the model to generate a machine-readable PSM, and (iii) scores the output with fuzzy structural and semantic metrics that reward partial but meaningful correctness. Figure 1 summarizes the pipeline.

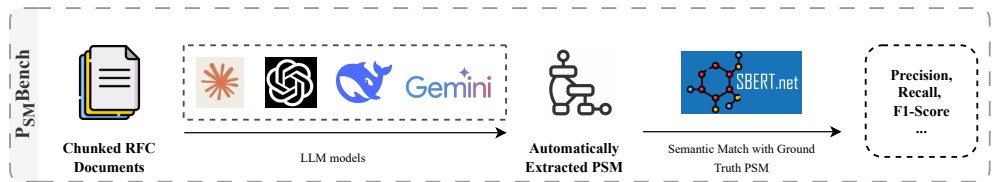

Figure 1: A workflow of PSMBENCH pipeline. The chunked RFC documents are input to LLM models to extract PSM automatically, and then the extracted PSM is compared with the ground truth PSM to evaluate completeness and correctness.

Our contributions are three-fold:

- **Comprehensive dataset.** RFC2PSM  is the first open-sourced dataset providing 1580 pages of cleaned RFC text, 108 manually domain-experts verified states, and 297 transitions covering 14 widely deployed protocols. This provides a dataset for LLMs on understanding network protocol technical documents.
- **New evaluation framework.** PSMBENCH introduces semantic-based fuzzy graph-alignment metrics that capture both exact matches and near-misses in extracted PSMs. It provides a benchmark for evaluating the LLM's capabilities in extracting complicated graph-like structures from technical documents.
- **Empirical baselines.** We benchmark 9 state-of-the-art open-source and commercial LLMs, highlighting persistent challenges such as long-range dependency tracking and state aliasing.

With this work, we aim to accelerate research in LLM-guided protocol analysis, ultimately contributing to the development of automated protocol analysis tools in the network and security community.

## 1.1  Background and Terminologies

**Network Protocols.** Network protocols define rules and conventions for data exchange across networks. For example, the Transmission Control Protocol (TCP) [Eddy, Wesley (Ed.), 2022] facilitates reliable internet communication, while the File Transfer Protocol (FTP) [Eddy, Wesley (Ed.), 2022] standardizes file exchanges. Given their widespread use in critical applications, the security and reliability of these protocols are paramount.

**Protocol Standards.** Network protocols are designed as interacting transition systems called Protocol State Machines (PSMs) and are specified in the Request for Comments (RFCs). These RFCs are official standards published and maintained by the Internet Engineering Task Force (IETF). Typically written in detailed natural language, RFC documents specify the technical aspects of protocol operation, including message formats, state transitions, and expected behaviors. Due to their complexity and extensive length, which often spans hundreds of pages, extracting structured information from RFCs poses significant challenges.

**Protocol State Machine (PSM)** A Protocol State Machine (PSM) provides a structured, formal representation of the states and transitions that define a network protocol's behavior in the form of finite state machine.

**Definition 1.1** (Protocol State Machine). *Formally, it's defined as a tuple [Brand and Zafiropulo, 1983]:*

$$\mathcal{M} = (S, \Sigma, T, s_0, F)$$

where $S$ is a finite set of states, representing all possible protocol states. $s_0 \in S$ is the initial state, where the protocol starts. $F \subseteq S$ is the set of final or terminal states, representing the valid end states of the protocol. $\Sigma$ is a finite set of events (or inputs) that can trigger state transitions. $T \subseteq S \times \Sigma \times S \times A$ is the set of transitions.

**Definition 1.2** (Transition). *A transition in a PSM is defined as a directed edge between two states, representing a valid state change triggered by an event. Formally, a transition $t$ is a 4-tuple: $t = (s_i, e, s_j, a)$, [Graham and Johnson, 2014] which can be represented as:*

$$s_i \xrightarrow{e/a} s_j$$

The transition indicates that the protocol can move from source state $s_i$ to destination state $s_j$ upon receiving the event $e$, with action $a$ being executed.

## 2   Related Work

**LLM-Based PSM Extraction.** Recent work has demonstrated that LLMs can automate the PSM extraction process. For example, Sharma et al. introduced PROSPER [Sharma and Yegneswaran, 2023], which uses GPT-3.5 with carefully engineered prompts to identify states and transitions directly from RFC text. Ma et al. [Ma et al., 2024] further explored LLM-guided extraction by selecting relevant sections of the IoT protocol specification [Connectivity Standards Alliance, 2022] and prompting an LLM to generate the corresponding state machine. However, their extraction approach is protocol-specific and is evaluated on only a handful of examples. In summary, although LLM-based techniques can reduce manual effort in PSM extraction, a major research gap is that there is no general, open-source, diverse benchmark or dataset against which to evaluate their performance across multiple protocols. Our work fills these gaps by introducing a unified dataset and benchmark covering 14 distinct protocols.

**Previously Proposed Benchmarks.** PROSPER [Sharma and Yegneswaran, 2023] is an attempt that explored PSM extraction from RFC documents. However, it has several limitations. It focuses on fewer than 10 protocols and evaluates a single LLM (GPT-3.5). Thus, its evaluation is limited. In contrast, our work introduces a substantially larger, publicly available dataset covering 14 distinct protocols, providing a more comprehensive and unified benchmark. Additionally, we evaluate 9 state-of-the-art LLM models, enabling systematic comparison and testing across a diverse range of protocols and models.

## 3   RFC2PSM  Dataset

We now introduce RFC2PSM , our comprehensive dataset designed to evaluate the ability of LLMs to extract complex protocol state machines (PSMs) from technical documents like RFCs. RFC2PSM covers 14 diverse protocols, each represented by a set of RFC document chunks as input source and manually annotated PSMs as the ground truth as shown in Figure 2. In this section, we detail the construction of RFC2PSM , including the protocol selection criteria, the RFC document preprocessing methods, and the structure of the ground truth PSMs.

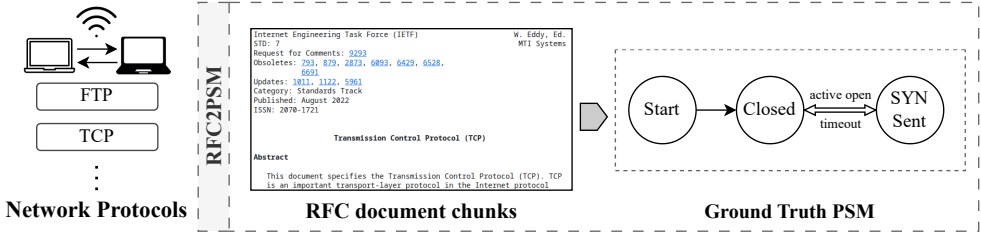

Figure 2: High-level overview of RFC2PSM , which includes 14 protocols, each protocol paired with RFC document chunks and ground truth PSM.

## 3.1 Protocol Selection and Dataset Statistics

We carefully select the protocols based on their widespread adoption and diverse coverage across different layers of the OSI (Open Systems Interconnection) model [iso, 1994], ensuring comprehensive representation across multiple communication contexts. RFC2PSM includes 14 protocols spanning the *Transport*, *Session*, *Application*, and *Data Link* layers, capturing a broad range of use cases. This diverse collection includes general-purpose network protocols (e.g., TCP, FTP), email protocols (e.g., SMTP, IMAP, POP3), real-time communication protocols (e.g., RTSP, SIP), routing protocols (e.g., BGP-4), and IoT-related protocols (e.g., MQTT). Collectively, the dataset encompasses **108** states, **297** transitions, and spans over **1580** pages of protocol specifications, reflecting the complexity and breadth of the selected standards. Detailed statistics, including protocol names, OSI layers, RFC standards, and other relevant information, are presented in Table 1 in the appendix.

## 3.2 RFC Document Collection and Preprocessing

For each protocol, we download the official RFC files, which serve as foundational references guiding real-world implementations. RFC documents are widely recognized within the security and network community as standards for protocol analysis tools, making them ideal sources for extracting ground truth PSMs in RFC2PSM .

Next, we outline the process of collecting and preprocessing these RFC documents:
① Each RFC document is first downloaded in plain text format from the official website of the Internet Engineering Task Force (IETF).
② The raw documents undergo a cleaning step to remove extra metadata, including page headers, footers, publication years, author information, and page numbers, ensuring that these elements do not introduce noise during LLM processing.
③ Given the token length limitations of LLMs, the cleaned RFCs are then segmented into structured, semantically coherent text chunks. This segmentation process first partitions the document at primary section boundaries (e.g., Sections 1, 2, 3). If a resulting chunk still exceeds the maximum token limit (e.g., 40,000 tokens), it is further divided at secondary-level subsections (e.g., Sections 1.1, 1.2).
④ During this segmentation, we retain section titles and numbers as explicit metadata, providing LLMs with crucial context for accurate PSM extraction.

As a result, each protocol's RFC is processed into a collection of structured chunks, where each chunk contains a *section identifier*, *section title*, and the corresponding *text content*, ensuring efficient and context-aware LLM processing.

## 3.3 Ground-Truth PSM

To evaluate the performance of PSM extraction, a well-defined ground-truth PSM is essential. In RFC2PSM , each network protocol is paired with a manually edited ground-truth PSM, capturing the valid protocol states and the events that trigger transitions between them.

To build reliable ground-truth PSMs, we first identify canonical state machines from existing trusted sources. Specifically, the PSM annotations for DCCP and TCP protocols are derived from prior work [Pacheco et al., 2022b] and have been manually validated for correctness. For the remaining protocols, we conduct a rigorous manual extraction process directly from the original RFC document. This manual step requires months of careful analysis and domain expertise to ensure the completeness

and accuracy of the resulting state machines. To make this process reproducible, we followed a systematic annotation protocol rather than relying on ad hoc effort. The annotators first established a concise two-page guideline covering state naming, event terminology, and action description. One author, acting as the protocol specialist, independently extracted all states and transitions from each RFC. A second author then reviewed the extracted PSM and marked revision points. Any differences were resolved through discussion until consensus was reached. On a 10% stratified sample, independent pass-1/2 annotations achieved substantial agreement ($\kappa$=0.82 for states and $\kappa$=0.78 for transitions) on Landis and Koch's scale, with fewer than 6% of elements requiring discussion in step 3. The full process averages about three days per protocol. The annotation guideline, reconciliation logs, and raw diff data will be released in the supplementary materials.

Each manually annotated PSM in RFC2PSM adheres to the formal definition presented in Definition 1.1. To ensure flexibility and ease of integration with existing tools, we represent each PSM as a structured JSON object, which allows flexible conversion to other widely adopted representations. Each graph-like PSM is structured as a JSON object comprising the elements in Definition 1.1. This structured format also facilitates the automatic generation of visual state diagrams, enhancing usability. For example, the ground-truth PSM for the TCP protocol is shown in Figure 5 in the appendix, illustrating the full set of protocol states and the transitions between them. In this representation, each transition is defined by a concise label of the form "trigger_event / action," capturing both the triggering condition and the resulting action.

## 4  PSMBENCH Benchmark

To systematically evaluate the ability of LLMs to understand and extract structured information from complex technical documents, we introduce PSMBENCH. In this section, we present the overall task definition and workflow for PSMBENCH, describing how LLMs extract PSMs from RFC chunks. We then provide a detailed explanation of the evaluation metrics used to assess the fidelity of these extracted PSMs, focusing on their ability to accurately capture the semantic relationships within the protocol's *states* and *transitions*.

**PSMBENCH Workflow.** The core task in PSMBENCH is to extract structured (PSMs) from RFC documents, as illustrated in Figure 1. The inputs are the chunked protocol's RFC sections. The goal is for the LLM to produce a structured, graph-like PSM in JSON format, capturing both *states* and *transitions* that accurately reflect the protocol's intended behavior. The extracted PSM is then compared with the ground-truth PSMs in RFC2PSM using semantic matching techniques, evaluating the model's ability to perform structured information extraction.

**Processing RFC Chunks with LLMs.** To extract a complete PSM from RFC documents, the LLM processes the segmented chunks, each corresponding to a distinct section of the document. For each chunk, the LLM identifies and extracts a *partial PSM* if it contains relevant protocol behavior information. Formally, a partial PSM is defined as a tuple $(S_{\text{partial}}, T_{\text{partial}})$, where $S_{\text{partial}}$ represents the set of protocol states mentioned in the section, and $T_{\text{partial}}$ captures the corresponding transitions, consistent with the transition Definition 1.2. Once all sections of an RFC document are processed, the LLM is prompted to merge the extracted partial PSMs to form a complete, global PSM. To assess whether segmentation granularity affects extraction quality, we further conducted a *sliding-window ablation*. Each RFC was re-segmented with a 4k-token window and 0.5k overlap, and the same extraction pipeline (GPT-4o-mini, identical hyper-parameters) was rerun. As detailed in Table 2 in the appendix, seven protocols improved and seven degraded, yielding a macro-average F1 change of +0.05, well within run-to-run variance. Because overlapping windows increase input length by approximately 35%, we retain section-based segmentation as the default for its determinism and efficiency.

### 4.1  Evaluation Metrics

The automated evaluation of PSM extraction is challenging due to the variability in phrases of semantically equivalent states and transitions. This variability makes the syntactic comparison inaccurate. In this subsection, we outline our approach to address this challenge through a semantic similarity-based evaluation. We then introduce the specific metrics used to quantify the fidelity of extracted *states* and *transitions*, providing a comprehensive framework for evaluating the structural and semantic alignment of LLM-generated PSMs.

### 4.1.1 Challenge and Solution in PSM Matching

**Challenge.** Evaluating the fidelity of extracted PSMs is particularly challenging due to the variability in *state names*, *events*, and *actions*. Direct string matching is often inaccurate, as semantically equivalent labels can have different lexical forms (e.g., *"Established"* vs. *"Connected"*), while superficially similar terms can have entirely distinct meanings (e.g., *"ACK"* and *"NACK"* represent opposite concepts). This variability complicates automated comparison, as minor differences in phrasing can significantly impact matching accuracy. Previous approaches [Pacheco et al., 2022b, Ma et al., 2024, Sharma and Yegneswaran, 2023] have often relied on manual evaluation, where experts align extracted PSMs with ground-truth references through detailed inspection. However, this approach is time-consuming and difficult to scale, presenting a major bottleneck in large-scale PSM evaluation. To address these challenges, automated metrics that capture semantic equivalence are essential for effective benchmarking.

**Solution.** We compute semantic similarity with sentence encoders, using `all-MiniLM-L6-v2` by default, and verified robustness across `all-MPNet-base-v2` and `SimCSE-RoBERTa-unsup`; macro F1 changes by at most $0.06$ with 11/14 protocols shifting by $\leq 0.15$ F1 (Table 4 in the appendix). This model generates dense, context-aware embeddings for each phrase, capturing their semantic relationships beyond syntax text similarity. Formally, the semantic similarity between two phrases $p_1$ and $p_2$ is defined as the cosine similarity of their SentenceBERT embeddings:

$$\text{sim}(p_1, p_2) = \frac{\mathbf{e}_1 \cdot \mathbf{e}_2}{\|\mathbf{e}_1\|\|\mathbf{e}_2\|} \tag{1}$$

where $\mathbf{e}_1 := \text{SentenceBERT}(p_1)$, $\mathbf{e}_2 := \text{SentenceBERT}(p_2)$, $\mathbf{e}_1 \cdot \mathbf{e}_2$ denotes the dot product, and $\|\mathbf{e}\|$ represents the Euclidean norm of the embedding vector. Two phrases are considered semantically equivalent if: $\text{sim}(p_1, p_2) > \theta$, where $\theta$ is a predefined threshold, selected based on empirical analysis (e.g., $\theta = 0.5$ in our experiments, we set it via an ablation study A). This threshold ensures that the metric captures meaningful semantic matches while filtering out lexical similarities.

### 4.1.2 State-Level Matching

Formally, given a ground truth PSM and an extracted PSM:

$$\mathcal{M} = (S, \Sigma, T, s_0, F) \quad \text{and} \quad \mathcal{M}' = (S', \Sigma', T', s_0', F')$$

**State Set Matching.** We assess the overall accuracy of state extraction, as states are fundamental to modeling protocol behavior. we aim to identify the overlap between the state sets $S$ and $S'$ of ground truth PSM and LLM extracted PSM. The states' similarity is calculated by $\text{sim}(s_i, s_j{'})$. A state is considered matched if the highest similarity score exceeds a threshold $\theta$ (e.g., 0.5). This approach captures both exact matches and semantically equivalent state names.

### 4.1.3 Transition-Level Matching

To assess the transition-level match, we define two kinds of matching.

**Exact Transition Match.** A transition is considered an exact match if all its components, including the *from state*, *to state*, *trigger event*, and *action*, are semantically equivalent to the corresponding transition in the ground truth PSM. Formally, a transition $t$ is considered a full match with transition $t'$ if: $t = (s_i, e, s_j, a)$ and $t' = (s_i', e', s_j', a')$ satisfy the following conditions:

$$\text{sim}(s_i, s_i') > \theta \wedge \text{sim}(s_j, s_j') > \theta \wedge \text{sim}(e \parallel a, e' \parallel a') > \theta$$

where $s_i$ and $s_i'$ are the source states, $s_j$ and $s_j'$ are the destination states, $e$ and $e'$ are the trigger events, and $a$ and $a'$ are the actions. The notation $e \parallel a$ represents the concatenation of the event and action, acting as a *transition label* in Definition 1.2.

**Partial Transition Match.** In cases where exact matching is too strict, we define a partial transition match based on semantic similarity, release the restriction of *event* or *action* descriptions. A partial match is considered valid if the semantic similarity of the source and destination states is above a predefined threshold, and at least one of the event or action components also satisfies the similarity requirement. Formally, a transition $t = (s_i, e, s_j, a)$ and $t' = (s_i', e', s_j', a')$ are considered a partial match if:

$$\text{sim}(s_i, s_i') > \theta \wedge \text{sim}(s_j, s_j') > \theta \wedge (\text{sim}(e, e') > \theta \vee \text{sim}(a, a') > \theta)$$

This approach reflects the practical observation that, while the fundamental state transitions remain consistent, the triggers and actions can be described with a wide range of context-dependent variations, making exact matching too restrictive.

### 4.1.4 Precision, Recall, and F1 Score for PSM Evaluation

To evaluate PSM extraction, we use precision, recall, and F1 score in a unified framework that can be applied to both *state*-level and *transition*-level matching. Let (Matched) be the set of elements that have been correctly identified by the extraction model; (Extracted) be the set of all elements produced by the LLM model; (Ground Truth) be the set of all reference elements from the ground-truth PSM. The evaluation metrics are then defined as:

$$\text{Precision} = \frac{|\text{Correct}|}{|\text{Extracted}|}, \quad \text{Recall} = \frac{|\text{Correct}|}{|\text{Ground Truth}|}, \quad \text{F1 Score} = 2 \times \frac{\text{Precision} \times \text{Recall}}{\text{Precision} + \text{Recall}}$$

In this context, *Precision* quantifies the proportion of correctly identified *states* or *transitions* among all extracted elements. *Recall* measures the fraction of ground-truth *states* or *transitions* that are successfully extracted. The *F1 Score* provides a balanced assessment, integrating both precision and recall into a single metric, to evaluate overall model performance.

## 5 Experiments

We conducted extensive baseline experiments to benchmark 9 state-of-the-art LLMs on the RFC2PSM dataset with PSMBENCH benchmark. In this section, we describe the selection of both open and proprietary LLMs, detail the prompt design, and present parameter settings. Finally, we provide a quantitative analysis of the results, offering insights into the strengths and limitations of current LLMs in the context of PSM extraction.

### 5.1 Experiments Setting

**Models.** i) Proprietary LLMs: For proprietary models, we evaluate several state-of-the-art LLMs with extensive context capabilities, including *Gpt4o-Mini* (gpt-4o-mini) [OpenAI, 2023], *Claude3* (claude-3-7-sonnet-20250219) [Anthropic, 2024], and *Gemini2* (gemini-2.0-flash) [DeepMind, 2024]. ii) Open LLMs: We also include a diverse set of advanced open-source instruction-tuned LLMs, including *DS-R1* (deepseek-R1) [DeepSeek-AI et al., 2025], *DS-V3* (deepseek-V3-0324), *QWQ* (qwq:32b), *QWen3* (qwen3:32b), *Gemma3* (gemma3:27b), and *Mistral* (mistral-small3.1:24b).

**Prompt Design.** To effectively guide LLMs in extracting PSMs from RFC documents, we adopt a two-stage prompt design inspired by the Chain of Thought (CoT) framework [Zhang et al., 2024]. Given the complexity and length of RFCs, we first segment each document into manageable sections to avoid token limits and ensure coherent extraction. This processing enables the LLM to focus on extracting partial PSM components from each section before combining them into a complete state machine. First, we use a *Partial PSM Extraction Prompt* (Appendix B.1) to extract PSM components (states and transitions) from individual sections. This step isolates meaningful state machine elements without overwhelming the model with the entire document context. Next, we use a *PSM Combination Prompt* (Appendix B.2) to merge these partial PSMs into a unified global PSM, ensuring consistency and completeness across sections.

**Parameter Settings.** For our experiments, we set the model *temperature = 0.0*, as the task of PSM extraction relies exclusively on the provided context. This deterministic setting ensures that the extracted state machines are consistent across runs, reflecting the contents of the RFC documents. For the semantic similarity threshold used in state and transition matching, we chose a value of *0.5*. We determined this threshold to balance the need for flexibility in matching semantically similar phrases while maintaining alignment with human interpretation. For instance, $sim(\texttt{Error}, \texttt{Failure}) = 0.5194$. From this example, we observe that a 0.5 threshold effectively captures meaningful semantic similarities without being overly strict.

### 5.2 Quantitative Results

In this subsection, we present the quantitative evaluation results for both *state*-level and *transition*-level matching, comparing LLM-extracted PSMs against their ground-truth counterparts. For each

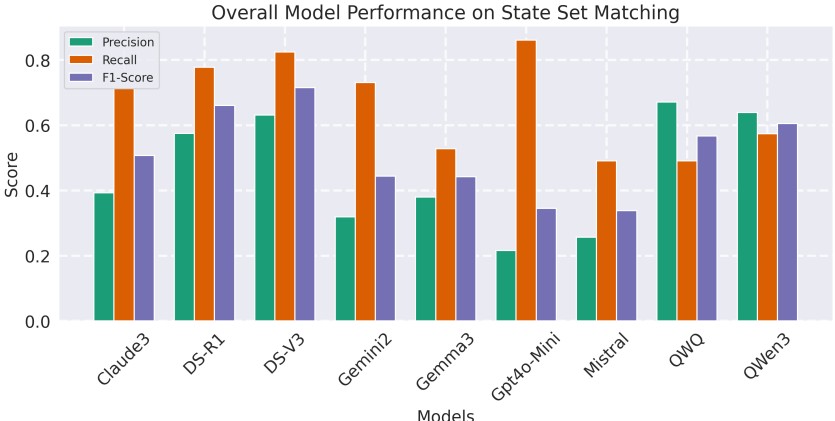

Figure 3: Model Performance on State Set Matching. Precision, recall, and F1-score for various models, highlighting differences in their ability to extract state sets accurately.

of the 14 protocols, we provide detailed performance tables covering *state set extraction*, *partial transition matching*, and *exact transition matching*. These tables are included in the Appendix for comprehensive reference, with an index to all detailed result tables provided in Table 5.

### 5.2.1 State-Level Matching Results

**States Set Matching Results.** Accurate state extraction is a critical component in reconstructing PSMs, as states define the fundamental stages of protocol behavior. In this evaluation, we measure the alignment between ground truth states ($S$) and extracted states ($S'$) based on semantic similarity, as described in the metrics section. Figure 3 presents the performance of each model in terms of total extracted states, ground truth states, matched states, and the corresponding precision, recall, and F1-score. Overall, *DS-V3* demonstrates the strongest performance, achieving the highest F1-score of 0.715, reflecting a balanced ability to capture both precise and diverse state representations. In contrast, models like *Gpt4o-Mini* and *Gemini2*, while achieving high recall (0.861 and 0.731, respectively), suffer from lower precision (0.216 and 0.319), indicating a tendency to over-extract states, possibly due to more aggressive token matching or broader semantic interpretations. On the other hand, *QWQ* and *QWen3* achieved high precision (0.671 and 0.639), but with a noticeable drop in recall, suggesting a more conservative approach to state identification that may miss relevant but less directly phrased states, the tradeoff between precision and recall is shown in Figure 4a. The numeric details are shown in Table 6 in the appendix.

### 5.2.2 Transition Level Matching Results

**Partial Transition Match Results.** Partial transition matching provides a more flexible evaluation approach, allowing for minor variations in event and action descriptions while still requiring strong alignment of source and destination states. Figure 6 in the appendix presents the partial transition matching results for each model. Overall, *DS-V3* achieves the highest F1-score (0.381), indicating a strong balance between precision and recall despite the relaxed matching criteria. This suggests that DS-V3 effectively captures the essential transition structures while accommodating variations in trigger and action descriptions. In contrast, models like *Gemini2* and *Gpt4o-Mini* demonstrate high recall (0.465 and 0.229, respectively) but suffer from significantly lower precision (0.138 and 0.101), indicating a tendency to over-generate transitions, possibly capturing many loosely related state changes. On the other hand, *QWQ* and *QWen3*, while achieving higher precision (0.239 and 0.288), exhibit lower recall, suggesting a more conservative extraction strategy that may miss relevant but less directly phrased transitions. The tradeoffs between recall and precision are shown in Figure 4b. These results highlight the challenges of extracting structured PSMs, where the complicated relationships between states, triggers, and actions require both precise matching and flexible interpretation. The detailed numeric results are shown in Table 7 in the appendix.

**Exact Transition Matching Results.** Exact transition matching provides a stricter evaluation, requiring precise alignment of source state, destination state, trigger event, and action. As shown in

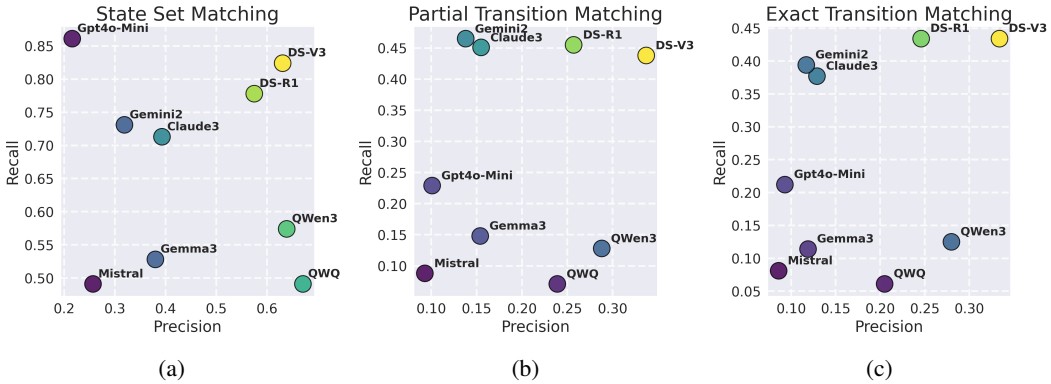

Figure 4: The figure presents the precision-recall distributions for (a) state set matching, (b) partial transition matching, and (c) exact transition matching, illustrating the varying performance of models across these metrics.

Figure 7 in the appendix, *DS-V3* achieved the highest F1-score (0.378), reflecting its ability to capture precise, context-aware transitions. In contrast, models like *Mistral* and *Gpt4o-Mini* struggled, with F1-scores of 0.083 and 0.130, respectively, indicating challenges in accurately aligning all transition components. These results underscore the difficulty of exact matching in PSM extraction, where even minor variations in state or event descriptions can significantly impact overall performance. The detailed numeric results are shown in Table 8 in the appendix.

## 5.3 Takeaways

The experimental results reveal several critical insights into the performance of state-of-the-art LLMs in extracting PSMs from RFC documents:

**State-Level Advantage in Extraction.** First, state-level extraction is generally more accurate than transition-level extraction across all models. This is evident from the higher F1-scores achieved in the *States Set Matching* task, where models like *DS-V3* (0.715) and *DS-R1* (0.661) significantly outperformed their transition-level counterparts, even under partially correct matching constraints. This suggests that capturing discrete, isolated states is a more straightforward task for LLMs than identifying the nuanced relationships represented by transitions, which involve multiple components.

**Impact of Model Scale.** Second, larger models with broader context capabilities, such as *DS-R1* and *DS-V3*, consistently outperformed smaller models like *QWQ* and *Mistral* across all metrics, including state and transition extraction matching. This highlights the advantage of large-scale models in handling the extensive, semantically complex inputs typical of RFC documents. However, these larger models also exhibited a tendency to over-extract, as reflected in their higher recall but lower precision, indicating potential challenges in accurately filtering relevant states and transitions.

**Challenges in Exact Transition Matching.** Third, the exact transition matching results underscore the difficulty of precise PSM extraction, where even the strongest models like *DS-V3* and *DS-R1* achieved only moderate F1-scores (0.378 and 0.314, respectively). This gap suggests that, despite their advanced reasoning capabilities, current LLMs struggle to consistently align all transition components accurately, reflecting the inherent complexity of protocol semantics.

## 6 Limitations and Future Work

While this work establishes a comprehensive benchmark PSMBENCH for PSM extraction from RFC documents, several limitations remain. First, our prompt design is straightforward, focusing on baseline evaluation without leveraging advanced strategies. This choice was made to provide a clear baseline, but it may limit the performance ceiling of some models. Second, while our dataset spans a diverse range of protocols, it primarily covers medium-sized, application-layer protocols. It excludes more complex, lower-layer protocols like Wi-Fi, whose specifications often exceed 1000 pages, posing significant challenges for current LLMs due to their extreme length and technical detail.

In future work, we plan to (i) extend RFC2PSM to ultra-long, lower-layer standards (e.g. Wi-Fi), (ii) explore adaptive chunk-merging and curriculum-style prompting to narrow the transition gap, and (iii) integrate richer graph-and-text co-evaluation metrics.

# 7  Conclusion

We present RFC2PSM , the first large-scale, manually validated corpus of 14 network-protocol specifications paired with ground-truth PSM, and PSMBENCH, a principled benchmark that evaluates LLM-driven PSM extraction through semantic, structure-aware metrics. Together they deliver a turn-key testbed for studying how current and future language models reason over long, technical documents and emit executable graph structures - an ability central to automated security analysis, software verification, and protocol testing.

Our extensive baseline study across 9 leading open-source and commercial LLMs reveals a clear state-transition gap: models achieve respectable recall on individual states yet struggle to assemble precise, end-to-end transition graphs. These findings pinpoint long-range dependency tracking, alias resolution, and fine-grained action/event disambiguation as open research challenges. By open-sourcing the dataset, evaluation code, and model outputs, we provide the community with a fully reproducible reference point and invite contributions ranging from prompt engineering and retrieval-augmented decoding to task-specific fine-tuning.

We hope this resource sparks broader collaboration across NLP, security, and networking, ultimately accelerating progress toward LLMs that can interpret and verify the protocols that run the Internet safely, accurately, and at scale.

## Acknowledgment

This work is supported by NSF Grant No. 2112471, the University of Texas System Rising STARs Award, and the startup funding from the University of Texas at Dallas.

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

Table 1: Overview of the dataset RFC2PSM .

| Protocol | Layer | RFC No. | # Pages | #Chunks | #States | #Transitions |
|---|---|---|---|---|---|---|
| RTSP | Application (L7) | RFC 7826 | 318 | 32 | 3 | 33 |
| FTP | Application (L7) | RFC 959 | 69 | 11 | 10 | 24 |
| SIP | Application (L7) | RFC 3261 | 269 | 31 | 5 | 20 |
| SMTP | Application (L7) | RFC 5321 | 95 | 15 | 7 | 22 |
| DCCP | Transport (L4) | RFC 4340 | 129 | 22 | 9 | 25 |
| TCP | Transport (L4) | RFC 9293 | 98 | 10 | 11 | 20 |
| DHCPv4 | Application (L7) | RFC 2131 | 45 | 9 | 8 | 19 |
| IMAP | Application (L7) | RFC 9051 | 163 | 11 | 5 | 11 |
| POP3 | Application (L7) | RFC 1939 | 23 | 17 | 3 | 18 |
| NNTP | Application (L7) | RFC 3977 | 125 | 14 | 10 | 16 |
| MQTT | Application (L7) | RFC 9431 | 33 | 11 | 12 | 17 |
| PPTP | Session (L5) | RFC 2637 | 57 | 8 | 9 | 19 |
| BGP-4 | Application (L7) | RFC 4271 | 104 | 16 | 6 | 26 |
| PPP | Data Link (L2) | RFC 1661 | 52 | 6 | 10 | 27 |

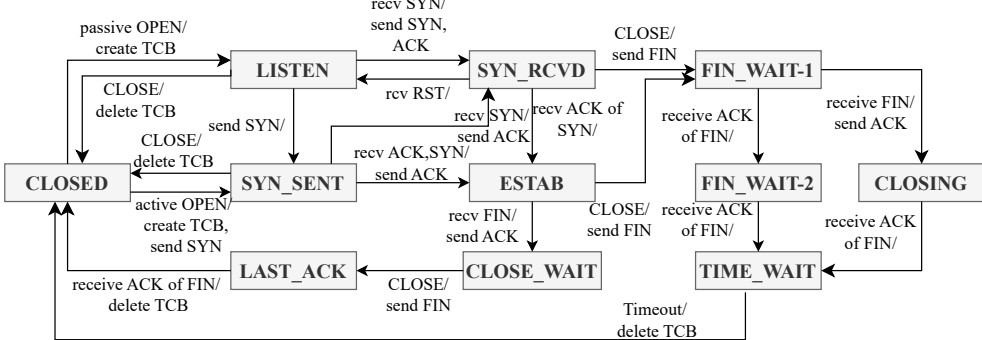

Figure 5: A manually extracted ground-truth PSM of TCP protocol in RFC2PSM .

# A   Threshold Ablation

When selecting the merge threshold, we systematically examined dozens of term pairs to balance false merges (distinct concepts collapsed) and false splits (identical concepts separated), and therefore set $\theta$=0.50. To make this choice explicit, Table 3 reports per–protocol F1 on the full `deepseek-v3` run (14 protocols, 297 transitions) for $\theta \in 0.40, 0.45, 0.50, 0.55, 0.60$.

These results confirm the choice: varying $\theta$ within $[0.40, 0.55]$ leaves macro F1 at 0.692, and even $\theta$=0.60 yields 0.664. At $\theta$=0.50, pairs such as "*Established* $\approx$ *Connected*" (sim = 0.77) remain cleanly separated from "*ACK* vs. *NACK*" (sim $< 0.2$), so we use $\theta$=0.50 as the default.

Table 2: Sliding-window ablation on state-matching F1. "Section" uses section-based chunks; "Overlap" uses a 4k window with 0.5k overlap.

| Protocol | Section F1 | Overlap F1 | $\Delta$ F1 |
|---|---|---|---|
| IMAP | 0.307 | 0.277 | $-0.030$ |
| POP3 | 0.353 | 1.000 | $+0.647$ |
| MQTT | 0.611 | 0.800 | $+0.189$ |
| PPP | 0.689 | 0.869 | $+0.180$ |
| PPTP | 0.414 | 0.485 | $+0.071$ |
| BGP | 0.632 | 0.800 | $+0.168$ |
| SIP | 0.064 | 0.070 | $+0.006$ |
| RTSP | 0.120 | 0.079 | $-0.041$ |
| DCCP | 0.361 | 0.361 | $+0.000$ |
| DHCP | 0.842 | 0.800 | $-0.042$ |
| FTP | 0.235 | 0.134 | $-0.101$ |
| NNTP | 0.400 | 0.328 | $-0.072$ |
| SMTP | 0.167 | 0.261 | $+0.094$ |
| TCP | 0.909 | 0.526 | $-0.383$ |
| **Macro Avg** | **0.436** | **0.485** | **+0.049** |

Table 3: Per–protocol F1 at different merge thresholds $\theta$. Macro average is flat for $\theta \in [0.40, 0.55]$ and drops slightly at $\theta=0.60$.

| Protocol | F1@0.40 | F1@0.45 | F1@0.50 | F1@0.55 | F1@0.60 |
|---|---|---|---|---|---|
| IMAP | 0.889 | 0.889 | 0.889 | 0.889 | 0.889 |
| POP3 | 0.750 | 0.750 | 0.750 | 0.750 | 0.750 |
| MQTT | 0.500 | 0.500 | 0.500 | 0.500 | 0.500 |
| PPP | 0.720 | 0.720 | 0.720 | 0.720 | 0.640 |
| PPTP | 0.477 | 0.477 | 0.477 | 0.477 | 0.477 |
| BGP | 1.000 | 1.000 | 1.000 | 1.000 | 1.000 |
| SIP | 0.589 | 0.589 | 0.589 | 0.589 | 0.589 |
| RTSP | 1.000 | 1.000 | 1.000 | 1.000 | 1.000 |
| DCCP | 0.857 | 0.857 | 0.857 | 0.857 | 0.857 |
| DHCP | 0.933 | 0.933 | 0.933 | 0.933 | 0.933 |
| FTP | 0.364 | 0.364 | 0.364 | 0.364 | 0.243 |
| NNTP | 0.444 | 0.444 | 0.444 | 0.444 | 0.370 |
| SMTP | 0.353 | 0.353 | 0.353 | 0.353 | 0.235 |
| TCP | 0.818 | 0.818 | 0.818 | 0.818 | 0.818 |
| **Macro avg.** | **0.692** | **0.692** | **0.692** | **0.692** | **0.664** |

Table 4: Per–protocol F1 with three sentence encoders (LLM, thresholds, and post–processing held fixed).

| Protocol | MiniLM F1 | MPNet F1 | SimCSE F1 |
|---|---|---|---|
| IMAP | 0.889 | 0.889 | 0.889 |
| POP3 | 0.750 | 0.750 | 0.750 |
| MQTT | 0.500 | 0.500 | 0.500 |
| PPP | 0.720 | 0.720 | 0.800 |
| PPTP | 0.477 | 0.477 | 0.667 |
| BGP | 1.000 | 1.000 | 1.000 |
| SIP | 0.589 | 0.589 | 0.589 |
| RTSP | 1.000 | 1.000 | 1.000 |
| DCCP | 0.857 | 0.857 | 0.857 |
| DHCP | 0.933 | 0.933 | 0.933 |
| FTP | 0.364 | 0.243 | 0.485 |
| NNTP | 0.444 | 0.444 | 0.519 |
| SMTP | 0.353 | 0.470 | 0.706 |
| TCP | 0.818 | 0.909 | 0.818 |
| Macro Avg. | 0.692 | 0.699 | 0.751 |

Table 5: References to Detailed Protocol Evaluation Tables

| Metric | BGP | FTP | IMAP | NNTP | POP3 | SMTP | SIP |
|---|---|---|---|---|---|---|---|
| States Extraction | Table 9 | Table 12 | Table 13 | Table 15 | Table 16 | Table 21 | Table 20 |
| Partial Transition | Table 23 | Table 26 | Table 27 | Table 29 | Table 30 | Table 35 | Table 34 |
| Exact Transition | Table 37 | Table 40 | Table 41 | Table 43 | Table 44 | Table 49 | Table 48 |
| Metric | TCP | DCCP | MQTT | PPTP | RTSP | DHCP | PPP |
| States Extraction | Table 22 | Table 10 | Table 14 | Table 18 | Table 19 | Table 11 | Table 17 |
| Partial Transition | Table 36 | Table 24 | Table 28 | Table 32 | Table 33 | Table 25 | Table 31 |
| Exact Transition | Table 50 | Table 38 | Table 42 | Table 46 | Table 47 | Table 39 | Table 45 |

Table 6: Overall Model Performance on **State Set Matching** of Different Protocols

| Model | Total Extracted | Total GT | Matched | Precision | Recall | F1-Score |
|---|---|---|---|---|---|---|
| Claude3 | 196 | 108 | 77 | 0.393 | 0.713 | 0.507 |
| DS-R1 | 146 | 108 | 84 | 0.575 | 0.778 | 0.661 |
| DS-V3 | 141 | 108 | 89 | 0.631 | 0.824 | 0.715 |
| Gemini2 | 248 | 108 | 79 | 0.319 | 0.731 | 0.444 |
| Gemma3 | 150 | 108 | 57 | 0.380 | 0.528 | 0.442 |
| Gpt4o-Mini | 431 | 108 | 93 | 0.216 | 0.861 | 0.345 |
| Mistral | 206 | 108 | 53 | 0.257 | 0.491 | 0.338 |
| QWQ | 79 | 108 | 53 | 0.671 | 0.491 | 0.567 |
| QWen3 | 97 | 108 | 62 | 0.639 | 0.574 | 0.605 |

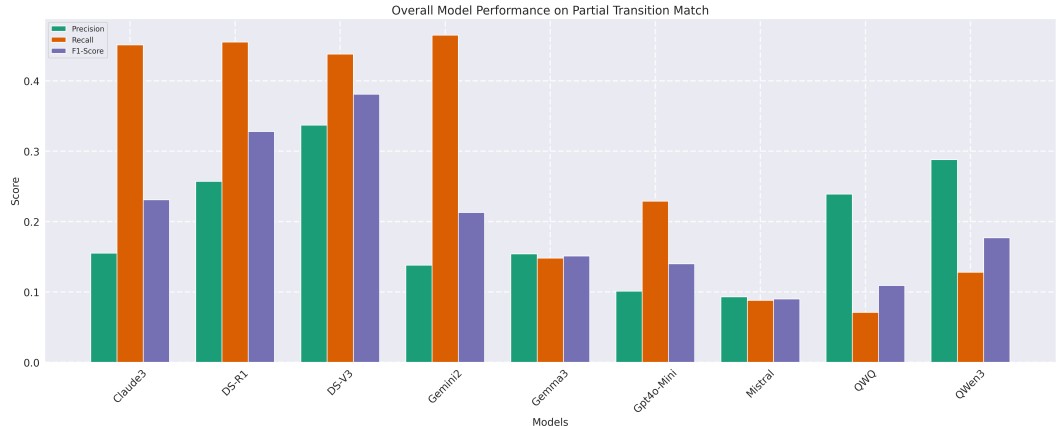

Figure 6: Model Performance on Partial Transition Matching. Precision, recall, and F1-score for various models, highlighting differences in their ability to extract transitions accurately.

Table 7: Overall Model Performance on **Partial Transition Match** of Different Protocols

| Model | TotalExtracted | TotalGT | Matched | Precision | Recall | F1-Score |
|---|---|---|---|---|---|---|
| Claude3 | 865 | 297 | 134 | 0.155 | 0.451 | 0.231 |
| DS-R1 | 525 | 297 | 135 | 0.257 | 0.455 | 0.328 |
| DS-V3 | 386 | 297 | 130 | 0.337 | 0.438 | 0.381 |
| Gemini2 | 1000 | 297 | 138 | 0.138 | 0.465 | 0.213 |
| Gemma3 | 285 | 297 | 44 | 0.154 | 0.148 | 0.151 |
| Gpt4o-Mini | 674 | 297 | 68 | 0.101 | 0.229 | 0.140 |
| Mistral | 279 | 297 | 26 | 0.093 | 0.088 | 0.090 |
| QWQ | 88 | 297 | 21 | 0.239 | 0.071 | 0.109 |
| QWen3 | 132 | 297 | 38 | 0.288 | 0.128 | 0.177 |

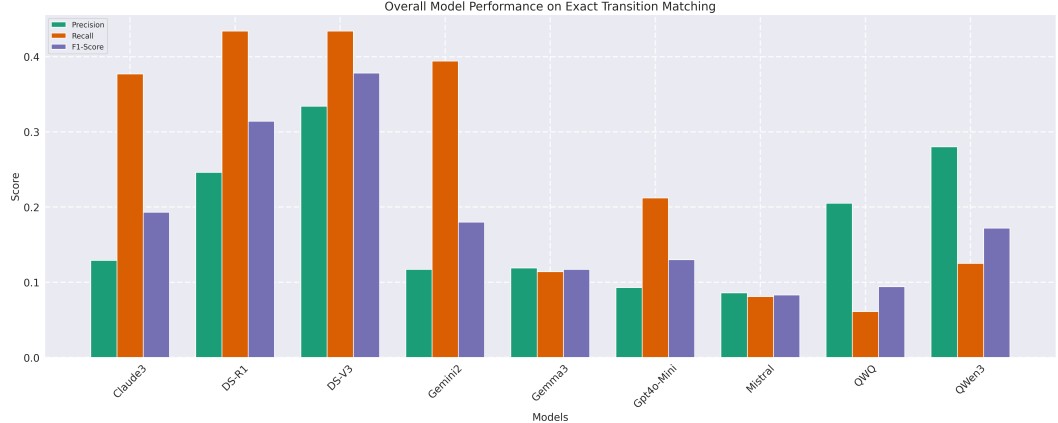

Figure 7: Model Performance on Exact Transition Matching. Precision, recall, and F1-score for various models, highlighting differences in their ability to extract transitions accurately.

Table 8: Overall Model Performance on Exact Transition Matching of Different Protocols

| Model | TotalExtracted | TotalGT | Matched | Precision | Recall | F1-Score |
|---|---|---|---|---|---|---|
| Claude3 | 865 | 297 | 112 | 0.129 | 0.377 | 0.193 |
| DS-R1 | 525 | 297 | 129 | 0.246 | 0.434 | 0.314 |
| DS-V3 | 386 | 297 | 129 | 0.334 | 0.434 | 0.378 |
| Gemini2 | 1000 | 297 | 117 | 0.117 | 0.394 | 0.180 |
| Gemma3 | 285 | 297 | 34 | 0.119 | 0.114 | 0.117 |
| Gpt4o-Mini | 674 | 297 | 63 | 0.093 | 0.212 | 0.130 |
| Mistral | 279 | 297 | 24 | 0.086 | 0.081 | 0.083 |
| QWQ | 88 | 297 | 18 | 0.205 | 0.061 | 0.094 |
| QWen3 | 132 | 297 | 37 | 0.280 | 0.125 | 0.172 |

# B   Prompts Design

## B.1   Partial PSM Extraction Prompt

---

**Partial PSM Extraction Prompt**

You will be given the section "section_title" of an RFC document for protocol "protocol_name".

**RESPONSE FORMAT (MANDATORY)**
- Your reply must consist **exclusively** of the JSON object representing the state machine.
- That JSON must be wrapped in <json> and </json> tags.
- Do **not** include any extra text, explanation, code fences, or formatting.
<section> section_text </section>
Steps:
1. Determine if this section has any FSM-related information (states, transitions, diagrams, reply codes, sequences).
2. If **none**, reply exactly:
<json>None</json>
3. If there is FSM information, extract it and return a structured JSON in the following format (strictly):

```
{{
   "states": ["state1", "state2", "state3"],
   "transitions": [
      {{
         "from": "state1",
         "event": "recvCommand",
         "action": "replCcode",
         "to": "state2"
      }},
      ...
   ]
}}
```

**FSM Field Constraints:**
"states"`:
List of all states appearing in '"from"'` or '"to"'` fields.
Each state must: Be 1 to 3 words (max 30 characters) Use 'CamelCase' or 'snake_case' Describe a protocol phase, status, or role (e.g., 'Authenticated', 'WaitingForReply') Contain no punctuation, spaces, or free-form descriptions
Good: '"AwaitingPassword"'`, '"transfer_in_progress"'`
Bad: '"State 1"'`, '"waiting for command"'`, '"cmd?"'`
"from" / "to":
- Same naming rules as above
"event":
- Describes the trigger that causes the transition
- Maybe begin with a fixed prefix:
- '"receive "'` for received command - '"send "'` for sent response - '"timeout "'` for timing event - '"cond "'` for internal condition - or other words
Examples: "receive USER", "send 230", "timeout 5s", "cond valid_credentials" '"action"':
- Describes what the system does in response
- It's best start with an action verb from this fixed set or other verbs if needed: 'reply', 'send', 'set', 'log', 'reset', 'close', 'collect', 'open', 'record', 'stop'
- Followed by one or two short arguments (max 4 words total) Examples: '"reply 230"'`, '"log failure"'`, '"set authenticated true"'`
Important:
- Do not generate free-text descriptions in any field.
- Each transition must contain **exactly**: 'from', 'event', 'action', 'to'.
- Do not invent vague or inconsistent state or event names.
**OUTPUT RULES:**
- Wrap the JSON in **<json>...</json>** only.
- Do not include Markdown, explanations, comments, or extra text.
- If nothing is found, return exactly '<json>None</json>'.

---

## B.2 PSM Combination Prompt

**Partial PSM Combination**

You will be provided with multiple \*\*partial protocol state machines\*\* extracted from different sections of an RFC. Each partial state machine is a JSON object with the following fields:
- "states": list of state names
- "transitions": list of transition objects with these required fields:
- "from": source state name
- "event": trigger (e.g., received command, condition)
- "action": response or internal action
- "to": target state name

Each partial is wrapped in '<partial>...</partial>'. Some may be '<json>None</json>' — ignore those.
— Your task is to \*\*merge all valid partial FSMs into one global FSM\*\* and return a single well-structured JSON object in the following format (wrapped in '<json>...</json>'):
<json>

```
{{
    "states": ["state1", "state2", ...],
    "initial_state": "stateX",
    "final_states": ["stateY", ...],
    "transitions": [
        {{
            "from": "state1",
            "event": "recv COMMAND",
            "action": "reply CODE",
            "to": "state2"
        }}
    ]
}}
```

</json>
— FSM Construction Constraints \*\*State Naming ('states', 'from', 'to')\*\*:
- Must be concise, meaningful, and consistent
- Format: 1 to 3 words, 'CamelCase' or 'snake_case', no spaces or punctuation
- Examples: '"Authenticated"', '"AwaitingPassword"', '"TransferReady"'
\*\*Events ('event')\*\*:
- Format: 1 to 3 words
- Maybe begin with: 'receive', 'send', 'timeout', or 'cond'
- Examples: '"receive USER"', '"timeout 10s"', '"cond valid_credentials"'
\*\*Actions ('action')\*\*:
- Start with a verb from this list: 'reply', 'send', 'set', 'log', 'reset', 'close', 'collect', 'open', 'record', 'stop' or other verbs if needed
Followed by a short phrase (less than 4 words)
- Examples: '"reply 230"', '"set authenticated true"', '"log failure"'

FSM Merging Rules
1. \*\*Unify states\*\*: Standardize naming (e.g., merge '"Init"' and '"Initialization"' into one state).
2. \*\*Remove duplicates\*\*: Transitions that differ only in phrasing should be merged.
3. \*\*Preserve meaning\*\*: If two similar states clearly serve different roles, retain both.
4. \*\*Determine\*\*:
- '"initial_state"': The state with \*\*no incoming transitions\*\*
- '"final_states"': All states with \*\*no outgoing transitions\*\*
Please return \*\*only\*\* the merged FSM in the required format, wrapped inside '<json>...</json>'. Do \*\*not\*\* include explanations, commentary, or Markdown.

Here are the partial FSMs to merge:
partials_block

Table 9: BGP Protocol States Extraction Metrics

| Protocol | Model | Total Extracted | Total GT | Matched | Precision | Recall | F1-Score |
|----------|-------|-----------------|----------|---------|-----------|--------|----------|
| BGP | DS-R1 | 6 | 6 | 6 | 1.000 | 1.000 | 1.000 |
| BGP | Gpt4o-Mini | 13 | 6 | 6 | 0.462 | 1.000 | 0.632 |
| BGP | Claude3 | 8 | 6 | 6 | 0.750 | 1.000 | 0.857 |
| BGP | Gemini2 | 9 | 6 | 6 | 0.667 | 1.000 | 0.800 |
| BGP | DS-V3 | 6 | 6 | 6 | 1.000 | 1.000 | 1.000 |
| BGP | QWQ | 6 | 6 | 6 | 1.000 | 1.000 | 1.000 |
| BGP | QWen3 | 6 | 6 | 6 | 1.000 | 1.000 | 1.000 |
| BGP | Gemma3 | 7 | 6 | 6 | 0.857 | 1.000 | 0.923 |
| BGP | Mistral | 10 | 6 | 3 | 0.300 | 0.500 | 0.375 |

Table 10: DCCP Protocol States Extraction Metrics

| Protocol | Model | Total Extracted | Total GT | Matched | Precision | Recall | F1-Score |
|----------|-------|-----------------|----------|---------|-----------|--------|----------|
| DCCP | DS-R1 | 12 | 9 | 9 | 0.750 | 1.000 | 0.857 |
| DCCP | Gpt4o-Mini | 41 | 9 | 9 | 0.220 | 1.000 | 0.361 |
| DCCP | Claude3 | 12 | 9 | 9 | 0.750 | 1.000 | 0.857 |
| DCCP | Gemini2 | 20 | 9 | 9 | 0.450 | 1.000 | 0.621 |
| DCCP | DS-V3 | 12 | 9 | 9 | 0.750 | 1.000 | 0.857 |
| DCCP | QWQ | 4 | 9 | 6 | 1.500 | 0.667 | 0.923 |
| DCCP | QWen3 | 11 | 9 | 8 | 0.727 | 0.889 | 0.800 |
| DCCP | Gemma3 | 10 | 9 | 7 | 0.700 | 0.778 | 0.737 |
| DCCP | Mistral | 10 | 9 | 2 | 0.200 | 0.222 | 0.210 |

Table 11: DHCP Protocol States Extraction Metrics

| Protocol | Model | Total Extracted | Total GT | Matched | Precision | Recall | F1-Score |
|----------|-------|-----------------|----------|---------|-----------|--------|----------|
| DHCP | DS-R1 | 8 | 8 | 8 | 1.000 | 1.000 | 1.000 |
| DHCP | Gpt4o-Mini | 11 | 8 | 8 | 0.727 | 1.000 | 0.842 |
| DHCP | Claude3 | 8 | 8 | 8 | 1.000 | 1.000 | 1.000 |
| DHCP | Gemini2 | 6 | 8 | 4 | 0.667 | 0.500 | 0.572 |
| DHCP | DS-V3 | 7 | 8 | 8 | 1.143 | 1.000 | 1.067 |
| DHCP | QWQ | 4 | 8 | 5 | 1.250 | 0.625 | 0.833 |
| DHCP | QWen3 | 4 | 8 | 5 | 1.250 | 0.625 | 0.833 |
| DHCP | Gemma3 | 4 | 8 | 5 | 1.250 | 0.625 | 0.833 |
| DHCP | Mistral | 3 | 8 | 4 | 1.333 | 0.500 | 0.727 |

Table 12: FTP Protocol States Extraction Metrics

| Protocol | Model | Total Extracted | Total GT | Matched | Precision | Recall | F1-Score |
|----------|-------|-----------------|----------|---------|-----------|--------|----------|
| FTP | DS-R1 | 14 | 10 | 5 | 0.357 | 0.500 | 0.417 |
| FTP | Gpt4o-Mini | 41 | 10 | 6 | 0.146 | 0.600 | 0.235 |
| FTP | Claude3 | 14 | 10 | 2 | 0.143 | 0.200 | 0.167 |
| FTP | Gemini2 | 15 | 10 | 6 | 0.400 | 0.600 | 0.480 |
| FTP | DS-V3 | 23 | 10 | 8 | 0.348 | 0.800 | 0.485 |
| FTP | QWQ | 6 | 10 | 5 | 0.833 | 0.500 | 0.625 |
| FTP | QWen3 | 11 | 10 | 2 | 0.182 | 0.200 | 0.191 |
| FTP | Gemma3 | 15 | 10 | 5 | 0.333 | 0.500 | 0.400 |
| FTP | Mistral | 20 | 10 | 4 | 0.200 | 0.400 | 0.267 |

Table 13: IMAP Protocol States Extraction Metrics

| Protocol | Model | Total Extracted | Total GT | Matched | Precision | Recall | F1-Score |
|---|---|---|---|---|---|---|---|
| IMAP | DS-R1 | 9 | 5 | 4 | 0.444 | 0.800 | 0.571 |
| IMAP | Gpt4o-Mini | 21 | 5 | 4 | 0.190 | 0.800 | 0.307 |
| IMAP | Claude3 | 11 | 5 | 4 | 0.364 | 0.800 | 0.500 |
| IMAP | Gemini2 | 12 | 5 | 4 | 0.333 | 0.800 | 0.470 |
| IMAP | DS-V3 | 4 | 5 | 4 | 1.000 | 0.800 | 0.889 |
| IMAP | QWQ | 4 | 5 | 4 | 1.000 | 0.800 | 0.889 |
| IMAP | QWen3 | 4 | 5 | 3 | 0.750 | 0.600 | 0.667 |
| IMAP | Gemma3 | 10 | 5 | 4 | 0.400 | 0.800 | 0.533 |
| IMAP | Mistral | 10 | 5 | 4 | 0.400 | 0.800 | 0.533 |

Table 14: MQTT Protocol States Extraction Metrics

| Protocol | Model | Total Extracted | Total GT | Matched | Precision | Recall | F1-Score |
|---|---|---|---|---|---|---|---|
| MQTT | DS-R1 | 7 | 12 | 8 | 1.143 | 0.667 | 0.842 |
| MQTT | Gpt4o-Mini | 24 | 12 | 11 | 0.458 | 0.917 | 0.611 |
| MQTT | Claude3 | 11 | 12 | 5 | 0.455 | 0.417 | 0.435 |
| MQTT | Gemini2 | 26 | 12 | 10 | 0.385 | 0.833 | 0.527 |
| MQTT | DS-V3 | 4 | 12 | 5 | 1.250 | 0.417 | 0.625 |
| MQTT | QWQ | 14 | 12 | 8 | 0.571 | 0.667 | 0.615 |
| MQTT | QWen3 | 7 | 12 | 5 | 0.714 | 0.417 | 0.527 |
| MQTT | Gemma3 | 11 | 12 | 5 | 0.455 | 0.417 | 0.435 |
| MQTT | Mistral | 20 | 12 | 11 | 0.550 | 0.917 | 0.688 |

Table 15: NNTP Protocol States Extraction Metrics

| Protocol | Model | Total Extracted | Total GT | Matched | Precision | Recall | F1-Score |
|---|---|---|---|---|---|---|---|
| NNTP | DS-R1 | 12 | 10 | 3 | 0.250 | 0.300 | 0.273 |
| NNTP | Gpt4o-Mini | 40 | 10 | 10 | 0.250 | 1.000 | 0.400 |
| NNTP | Claude3 | 13 | 10 | 6 | 0.462 | 0.600 | 0.522 |
| NNTP | Gemini2 | 29 | 10 | 7 | 0.241 | 0.700 | 0.359 |
| NNTP | DS-V3 | 17 | 10 | 7 | 0.412 | 0.700 | 0.519 |
| NNTP | QWQ | 3 | 10 | 0 | 0.000 | 0.000 | 0.000 |
| NNTP | QWen3 | 4 | 10 | 0 | 0.000 | 0.000 | 0.000 |
| NNTP | Gemma3 | 9 | 10 | 3 | 0.333 | 0.300 | 0.316 |
| NNTP | Mistral | 3 | 10 | 0 | 0.000 | 0.000 | 0.000 |

Table 16: POP3 Protocol States Extraction Metrics

| Protocol | Model | Total Extracted | Total GT | Matched | Precision | Recall | F1-Score |
|---|---|---|---|---|---|---|---|
| POP3 | DS-R1 | 4 | 3 | 3 | 0.750 | 1.000 | 0.857 |
| POP3 | Gpt4o-Mini | 14 | 3 | 3 | 0.214 | 1.000 | 0.353 |
| POP3 | Claude3 | 5 | 3 | 3 | 0.600 | 1.000 | 0.750 |
| POP3 | Gemini2 | 4 | 3 | 3 | 0.750 | 1.000 | 0.857 |
| POP3 | DS-V3 | 5 | 3 | 3 | 0.600 | 1.000 | 0.750 |
| POP3 | QWQ | 4 | 3 | 1 | 0.250 | 0.333 | 0.286 |
| POP3 | QWen3 | 4 | 3 | 3 | 0.750 | 1.000 | 0.857 |
| POP3 | Gemma3 | 17 | 3 | 3 | 0.176 | 1.000 | 0.299 |
| POP3 | Mistral | 12 | 3 | 3 | 0.250 | 1.000 | 0.400 |

Table 17: PPP Protocol States Extraction Metrics

| Protocol | Model | Total Extracted | Total GT | Matched | Precision | Recall | F1-Score |
|---|---|---|---|---|---|---|---|
| PPP | DS-R1 | 15 | 10 | 10 | 0.667 | 1.000 | 0.800 |
| PPP | Gpt4o-Mini | 19 | 10 | 10 | 0.526 | 1.000 | 0.689 |
| PPP | Claude3 | 15 | 10 | 10 | 0.667 | 1.000 | 0.800 |
| PPP | Gemini2 | 13 | 10 | 10 | 0.769 | 1.000 | 0.869 |
| PPP | DS-V3 | 15 | 10 | 10 | 0.667 | 1.000 | 0.800 |
| PPP | QWQ | 5 | 10 | 3 | 0.600 | 0.300 | 0.400 |
| PPP | QWen3 | 11 | 10 | 6 | 0.545 | 0.600 | 0.571 |
| PPP | Gemma3 | 6 | 10 | 2 | 0.333 | 0.200 | 0.250 |
| PPP | Mistral | 9 | 10 | 2 | 0.222 | 0.200 | 0.210 |

Table 18: PPTP Protocol States Extraction Metrics

| Protocol | Model | Total Extracted | Total GT | Matched | Precision | Recall | F1-Score |
|---|---|---|---|---|---|---|---|
| PPTP | DS-R1 | 8 | 9 | 8 | 1.000 | 0.889 | 0.941 |
| PPTP | Gpt4o-Mini | 20 | 9 | 6 | 0.300 | 0.667 | 0.414 |
| PPTP | Claude3 | 15 | 9 | 5 | 0.333 | 0.556 | 0.417 |
| PPTP | Gemini2 | 11 | 9 | 0 | 0.000 | 0.000 | 0.000 |
| PPTP | DS-V3 | 12 | 9 | 8 | 0.667 | 0.889 | 0.762 |
| PPTP | QWQ | 3 | 9 | 5 | 1.667 | 0.556 | 0.834 |
| PPTP | QWen3 | 4 | 9 | 8 | 2.000 | 0.889 | 1.231 |
| PPTP | Gemma3 | 4 | 9 | 6 | 1.500 | 0.667 | 0.923 |
| PPTP | Mistral | 6 | 9 | 6 | 1.000 | 0.667 | 0.800 |

Table 19: RTSP Protocol States Extraction Metrics

| Protocol | Model | Total Extracted | Total GT | Matched | Precision | Recall | F1-Score |
|---|---|---|---|---|---|---|---|
| RTSP | DS-R1 | 5 | 3 | 3 | 0.600 | 1.000 | 0.750 |
| RTSP | Gpt4o-Mini | 47 | 3 | 3 | 0.064 | 1.000 | 0.120 |
| RTSP | Claude3 | 20 | 3 | 2 | 0.100 | 0.667 | 0.174 |
| RTSP | Gemini2 | 17 | 3 | 2 | 0.118 | 0.667 | 0.201 |
| RTSP | DS-V3 | 3 | 3 | 3 | 1.000 | 1.000 | 1.000 |
| RTSP | QWQ | 3 | 3 | 0 | 0.000 | 0.000 | 0.000 |
| RTSP | QWen3 | 3 | 3 | 3 | 1.000 | 1.000 | 1.000 |
| RTSP | Gemma3 | 14 | 3 | 3 | 0.214 | 1.000 | 0.353 |
| RTSP | Mistral | 22 | 3 | 1 | 0.045 | 0.333 | 0.079 |

Table 20: SIP Protocol States Extraction Metrics

| Protocol | Model | Total Extracted | Total GT | Matched | Precision | Recall | F1-Score |
|---|---|---|---|---|---|---|---|
| SIP | DS-R1 | 25 | 5 | 5 | 0.200 | 1.000 | 0.333 |
| SIP | Gpt4o-Mini | 88 | 5 | 3 | 0.034 | 0.600 | 0.064 |
| SIP | Claude3 | 43 | 5 | 5 | 0.116 | 1.000 | 0.208 |
| SIP | Gemini2 | 65 | 5 | 5 | 0.077 | 1.000 | 0.143 |
| SIP | DS-V3 | 12 | 5 | 5 | 0.417 | 1.000 | 0.589 |
| SIP | QWQ | 7 | 5 | 0 | 0.000 | 0.000 | 0.000 |
| SIP | QWen3 | 11 | 5 | 2 | 0.182 | 0.400 | 0.250 |
| SIP | Gemma3 | 27 | 5 | 2 | 0.074 | 0.400 | 0.125 |
| SIP | Mistral | 32 | 5 | 1 | 0.031 | 0.200 | 0.054 |

Table 21: SMTP Protocol States Extraction Metrics

| Protocol | Model | Total Extracted | Total GT | Matched | Precision | Recall | F1-Score |
|---|---|---|---|---|---|---|---|
| SMTP | DS-R1 | 10 | 7 | 2 | 0.200 | 0.286 | 0.235 |
| SMTP | Gpt4o-Mini | 41 | 7 | 4 | 0.098 | 0.571 | 0.167 |
| SMTP | Claude3 | 10 | 7 | 2 | 0.200 | 0.286 | 0.235 |
| SMTP | Gemini2 | 10 | 7 | 4 | 0.400 | 0.571 | 0.470 |
| SMTP | DS-V3 | 10 | 7 | 3 | 0.300 | 0.429 | 0.353 |
| SMTP | QWQ | 7 | 7 | 1 | 0.143 | 0.143 | 0.143 |
| SMTP | QWen3 | 8 | 7 | 2 | 0.250 | 0.286 | 0.267 |
| SMTP | Gemma3 | 7 | 7 | 0 | 0.000 | 0.000 | 0.000 |
| SMTP | Mistral | 38 | 7 | 3 | 0.079 | 0.429 | 0.133 |

Table 22: TCP Protocol States Extraction Metrics

| Protocol | Model | Total Extracted | Total GT | Matched | Precision | Recall | F1-Score |
|---|---|---|---|---|---|---|---|
| TCP | DS-R1 | 11 | 11 | 10 | 0.909 | 0.909 | 0.909 |
| TCP | Gpt4o-Mini | 11 | 11 | 10 | 0.909 | 0.909 | 0.909 |
| TCP | Claude3 | 11 | 11 | 10 | 0.909 | 0.909 | 0.909 |
| TCP | Gemini2 | 11 | 11 | 9 | 0.818 | 0.818 | 0.818 |
| TCP | DS-V3 | 11 | 11 | 10 | 0.909 | 0.909 | 0.909 |
| TCP | QWQ | 9 | 11 | 9 | 1.000 | 0.818 | 0.900 |
| TCP | QWen3 | 9 | 11 | 9 | 1.000 | 0.818 | 0.900 |
| TCP | Gemma3 | 9 | 11 | 6 | 0.667 | 0.545 | 0.600 |
| TCP | Mistral | 11 | 11 | 9 | 0.818 | 0.818 | 0.818 |

Table 23: BGP Partially Correct Transition Extraction Metrics

| Protocol | Model | TotalExtracted | TotalGT | Matched | Precision | Recall | F1-Score |
|---|---|---|---|---|---|---|---|
| BGP | DS-R1 | 20 | 26 | 14 | 0.700 | 0.538 | 0.609 |
| BGP | Gpt4o-Mini | 25 | 26 | 7 | 0.280 | 0.269 | 0.275 |
| BGP | Claude3 | 60 | 26 | 23 | 0.383 | 0.885 | 0.535 |
| BGP | Gemini2 | 101 | 26 | 26 | 0.257 | 1.000 | 0.409 |
| BGP | DS-V3 | 32 | 26 | 15 | 0.469 | 0.577 | 0.517 |
| BGP | QWQ | 7 | 26 | 3 | 0.429 | 0.115 | 0.182 |
| BGP | QWen3 | 11 | 26 | 4 | 0.364 | 0.154 | 0.216 |
| BGP | Gemma3 | 21 | 26 | 3 | 0.143 | 0.115 | 0.128 |
| BGP | Mistral | 9 | 26 | 0 | 0.000 | 0.000 | 0.000 |

Table 24: DCCP Partially Correct Transition Extraction Metrics

| Protocol | Model | TotalExtracted | TotalGT | Matched | Precision | Recall | F1-Score |
|---|---|---|---|---|---|---|---|
| DCCP | DS-R1 | 29 | 25 | 18 | 0.621 | 0.720 | 0.667 |
| DCCP | Gpt4o-Mini | 55 | 25 | 8 | 0.145 | 0.320 | 0.200 |
| DCCP | Claude3 | 40 | 25 | 19 | 0.475 | 0.760 | 0.585 |
| DCCP | Gemini2 | 44 | 25 | 16 | 0.364 | 0.640 | 0.464 |
| DCCP | DS-V3 | 28 | 25 | 14 | 0.500 | 0.560 | 0.528 |
| DCCP | QWQ | 4 | 25 | 1 | 0.250 | 0.040 | 0.069 |
| DCCP | QWen3 | 15 | 25 | 5 | 0.333 | 0.200 | 0.250 |
| DCCP | Gemma3 | 18 | 25 | 6 | 0.333 | 0.240 | 0.279 |
| DCCP | Mistral | 9 | 25 | 0 | 0.000 | 0.000 | 0.000 |

Table 25: DHCP Partially Correct Transition Extraction Metrics

| Protocol | Model | TotalExtracted | TotalGT | Matched | Precision | Recall | F1-Score |
|---|---|---|---|---|---|---|---|
| DHCP | DS-R1 | 20 | 19 | 12 | 0.600 | 0.632 | 0.615 |
| DHCP | Gpt4o-Mini | 26 | 19 | 9 | 0.346 | 0.474 | 0.400 |
| DHCP | Claude3 | 18 | 19 | 14 | 0.778 | 0.737 | 0.757 |
| DHCP | Gemini2 | 27 | 19 | 7 | 0.259 | 0.368 | 0.304 |
| DHCP | DS-V3 | 15 | 19 | 10 | 0.667 | 0.526 | 0.588 |
| DHCP | QWQ | 6 | 19 | 4 | 0.667 | 0.211 | 0.320 |
| DHCP | QWen3 | 4 | 19 | 2 | 0.500 | 0.105 | 0.174 |
| DHCP | Gemma3 | 10 | 19 | 2 | 0.200 | 0.105 | 0.138 |
| DHCP | Mistral | 5 | 19 | 2 | 0.400 | 0.105 | 0.167 |

Table 26: FTP Partially Correct Transition Extraction Metrics

| Protocol | Model | TotalExtracted | TotalGT | Matched | Precision | Recall | F1-Score |
|---|---|---|---|---|---|---|---|
| FTP | DS-R1 | 23 | 24 | 5 | 0.217 | 0.208 | 0.213 |
| FTP | Gpt4o-Mini | 53 | 24 | 4 | 0.075 | 0.167 | 0.104 |
| FTP | Claude3 | 46 | 24 | 4 | 0.087 | 0.167 | 0.114 |
| FTP | Gemini2 | 51 | 24 | 6 | 0.118 | 0.250 | 0.160 |
| FTP | DS-V3 | 77 | 24 | 7 | 0.091 | 0.292 | 0.139 |
| FTP | QWQ | 7 | 24 | 3 | 0.429 | 0.125 | 0.194 |
| FTP | QWen3 | 14 | 24 | 1 | 0.071 | 0.042 | 0.053 |
| FTP | Gemma3 | 42 | 24 | 5 | 0.119 | 0.208 | 0.152 |
| FTP | Mistral | 33 | 24 | 3 | 0.091 | 0.125 | 0.105 |

Table 27: IMAP Partially Correct Transition Extraction Metrics

| Protocol | Model | TotalExtracted | TotalGT | Matched | Precision | Recall | F1-Score |
|---|---|---|---|---|---|---|---|
| IMAP | DS-R1 | 35 | 11 | 6 | 0.171 | 0.545 | 0.261 |
| IMAP | Gpt4o-Mini | 56 | 11 | 5 | 0.089 | 0.455 | 0.149 |
| IMAP | Claude3 | 48 | 11 | 7 | 0.146 | 0.636 | 0.237 |
| IMAP | Gemini2 | 90 | 11 | 7 | 0.078 | 0.636 | 0.139 |
| IMAP | DS-V3 | 16 | 11 | 6 | 0.375 | 0.545 | 0.444 |
| IMAP | QWQ | 4 | 11 | 3 | 0.750 | 0.273 | 0.400 |
| IMAP | QWen3 | 4 | 11 | 2 | 0.500 | 0.182 | 0.267 |
| IMAP | Gemma3 | 17 | 11 | 5 | 0.294 | 0.455 | 0.357 |
| IMAP | Mistral | 20 | 11 | 6 | 0.300 | 0.545 | 0.387 |

Table 28: MQTT Partially Correct Transition Extraction Metrics

| Protocol | Model | TotalExtracted | TotalGT | Matched | Precision | Recall | F1-Score |
|---|---|---|---|---|---|---|---|
| MQTT | DS-R1 | 16 | 17 | 2 | 0.125 | 0.118 | 0.121 |
| MQTT | Gpt4o-Mini | 37 | 17 | 3 | 0.081 | 0.176 | 0.111 |
| MQTT | Claude3 | 45 | 17 | 2 | 0.044 | 0.118 | 0.065 |
| MQTT | Gemini2 | 48 | 17 | 2 | 0.042 | 0.118 | 0.062 |
| MQTT | DS-V3 | 6 | 17 | 0 | 0.000 | 0.000 | 0.000 |
| MQTT | QWQ | 18 | 17 | 1 | 0.056 | 0.059 | 0.057 |
| MQTT | QWen3 | 6 | 17 | 1 | 0.167 | 0.059 | 0.087 |
| MQTT | Gemma3 | 19 | 17 | 1 | 0.053 | 0.059 | 0.056 |
| MQTT | Mistral | 37 | 17 | 3 | 0.081 | 0.176 | 0.111 |

Table 29: NNTP Partially Correct Transition Extraction Metrics

| Protocol | Model | TotalExtracted | TotalGT | Matched | Precision | Recall | F1-Score |
|----------|-------|----------------|---------|---------|-----------|--------|----------|
| NNTP | DS-R1 | 73 | 16 | 2 | 0.027 | 0.125 | 0.045 |
| NNTP | Gpt4o-Mini | 68 | 16 | 0 | 0.000 | 0.000 | 0.000 |
| NNTP | Claude3 | 75 | 16 | 0 | 0.000 | 0.000 | 0.000 |
| NNTP | Gemini2 | 116 | 16 | 2 | 0.017 | 0.125 | 0.030 |
| NNTP | DS-V3 | 25 | 16 | 2 | 0.080 | 0.125 | 0.098 |
| NNTP | QWQ | 3 | 16 | 0 | 0.000 | 0.000 | 0.000 |
| NNTP | QWen3 | 6 | 16 | 0 | 0.000 | 0.000 | 0.000 |
| NNTP | Gemma3 | 14 | 16 | 0 | 0.000 | 0.000 | 0.000 |
| NNTP | Mistral | 4 | 16 | 0 | 0.000 | 0.000 | 0.000 |

Table 30: POP3 Partially Correct Transition Extraction Metrics

| Protocol | Model | TotalExtracted | TotalGT | Matched | Precision | Recall | F1-Score |
|----------|-------|----------------|---------|---------|-----------|--------|----------|
| POP3 | DS-R1 | 21 | 18 | 12 | 0.571 | 0.667 | 0.615 |
| POP3 | Gpt4o-Mini | 33 | 18 | 12 | 0.364 | 0.667 | 0.471 |
| POP3 | Claude3 | 26 | 18 | 9 | 0.346 | 0.500 | 0.409 |
| POP3 | Gemini2 | 39 | 18 | 13 | 0.333 | 0.722 | 0.456 |
| POP3 | DS-V3 | 20 | 18 | 10 | 0.500 | 0.556 | 0.526 |
| POP3 | QWQ | 5 | 18 | 1 | 0.200 | 0.056 | 0.087 |
| POP3 | QWen3 | 8 | 18 | 4 | 0.500 | 0.222 | 0.308 |
| POP3 | Gemma3 | 38 | 18 | 10 | 0.263 | 0.556 | 0.357 |
| POP3 | Mistral | 29 | 18 | 9 | 0.310 | 0.500 | 0.383 |

Table 31: PPP Partially Correct Transition Extraction Metrics

| Protocol | Model | TotalExtracted | TotalGT | Matched | Precision | Recall | F1-Score |
|----------|-------|----------------|---------|---------|-----------|--------|----------|
| PPP | DS-R1 | 111 | 27 | 10 | 0.090 | 0.370 | 0.145 |
| PPP | Gpt4o-Mini | 33 | 27 | 5 | 0.152 | 0.185 | 0.167 |
| PPP | Claude3 | 138 | 27 | 18 | 0.130 | 0.667 | 0.218 |
| PPP | Gemini2 | 126 | 27 | 20 | 0.159 | 0.741 | 0.261 |
| PPP | DS-V3 | 21 | 27 | 3 | 0.143 | 0.111 | 0.125 |
| PPP | QWQ | 5 | 27 | 0 | 0.000 | 0.000 | 0.000 |
| PPP | QWen3 | 21 | 27 | 1 | 0.048 | 0.037 | 0.042 |
| PPP | Gemma3 | 9 | 27 | 0 | 0.000 | 0.000 | 0.000 |
| PPP | Mistral | 12 | 27 | 0 | 0.000 | 0.000 | 0.000 |

Table 32: PPTP Partially Correct Transition Extraction Metrics

| Protocol | Model | TotalExtracted | TotalGT | Matched | Precision | Recall | F1-Score |
|----------|-------|----------------|---------|---------|-----------|--------|----------|
| PPTP | DS-R1 | 18 | 19 | 12 | 0.667 | 0.632 | 0.649 |
| PPTP | Gpt4o-Mini | 29 | 19 | 1 | 0.034 | 0.053 | 0.042 |
| PPTP | Claude3 | 56 | 19 | 4 | 0.071 | 0.211 | 0.107 |
| PPTP | Gemini2 | 23 | 19 | 0 | 0.000 | 0.000 | 0.000 |
| PPTP | DS-V3 | 34 | 19 | 11 | 0.324 | 0.579 | 0.415 |
| PPTP | QWQ | 2 | 19 | 0 | 0.000 | 0.000 | 0.000 |
| PPTP | QWen3 | 6 | 19 | 4 | 0.667 | 0.211 | 0.320 |
| PPTP | Gemma3 | 6 | 19 | 2 | 0.333 | 0.105 | 0.160 |
| PPTP | Mistral | 12 | 19 | 1 | 0.083 | 0.053 | 0.065 |

Table 33: RTSP Partially Correct Transition Extraction Metrics

| Protocol | Model | TotalExtracted | TotalGT | Matched | Precision | Recall | F1-Score |
|---|---|---|---|---|---|---|---|
| RTSP | DS-R1 | 41 | 33 | 19 | 0.463 | 0.576 | 0.514 |
| RTSP | Gpt4o-Mini | 70 | 33 | 0 | 0.000 | 0.000 | 0.000 |
| RTSP | Claude3 | 89 | 33 | 19 | 0.213 | 0.576 | 0.311 |
| RTSP | Gemini2 | 78 | 33 | 12 | 0.154 | 0.364 | 0.216 |
| RTSP | DS-V3 | 39 | 33 | 30 | 0.769 | 0.909 | 0.833 |
| RTSP | QWQ | 2 | 33 | 0 | 0.000 | 0.000 | 0.000 |
| RTSP | QWen3 | 10 | 33 | 6 | 0.600 | 0.182 | 0.279 |
| RTSP | Gemma3 | 24 | 33 | 4 | 0.167 | 0.121 | 0.140 |
| RTSP | Mistral | 33 | 33 | 0 | 0.000 | 0.000 | 0.000 |

Table 34: SIP Partially Correct Transition Extraction Metrics

| Protocol | Model | TotalExtracted | TotalGT | Matched | Precision | Recall | F1-Score |
|---|---|---|---|---|---|---|---|
| SIP | DS-R1 | 58 | 20 | 12 | 0.207 | 0.600 | 0.308 |
| SIP | Gpt4o-Mini | 127 | 20 | 10 | 0.079 | 0.500 | 0.136 |
| SIP | Claude3 | 164 | 20 | 2 | 0.012 | 0.100 | 0.022 |
| SIP | Gemini2 | 180 | 20 | 20 | 0.111 | 1.000 | 0.200 |
| SIP | DS-V3 | 43 | 20 | 14 | 0.326 | 0.700 | 0.444 |
| SIP | QWQ | 5 | 20 | 0 | 0.000 | 0.000 | 0.000 |
| SIP | QWen3 | 11 | 20 | 3 | 0.273 | 0.150 | 0.194 |
| SIP | Gemma3 | 47 | 20 | 2 | 0.043 | 0.100 | 0.060 |
| SIP | Mistral | 30 | 20 | 0 | 0.000 | 0.000 | 0.000 |

Table 35: SMTP Partially Correct Transition Extraction Metrics

| Protocol | Model | TotalExtracted | TotalGT | Matched | Precision | Recall | F1-Score |
|---|---|---|---|---|---|---|---|
| SMTP | DS-R1 | 45 | 22 | 0 | 0.000 | 0.000 | 0.000 |
| SMTP | Gpt4o-Mini | 52 | 22 | 0 | 0.000 | 0.000 | 0.000 |
| SMTP | Claude3 | 40 | 22 | 0 | 0.000 | 0.000 | 0.000 |
| SMTP | Gemini2 | 32 | 22 | 2 | 0.062 | 0.091 | 0.074 |
| SMTP | DS-V3 | 18 | 22 | 1 | 0.056 | 0.045 | 0.050 |
| SMTP | QWQ | 11 | 22 | 0 | 0.000 | 0.000 | 0.000 |
| SMTP | QWen3 | 7 | 22 | 0 | 0.000 | 0.000 | 0.000 |
| SMTP | Gemma3 | 6 | 22 | 0 | 0.000 | 0.000 | 0.000 |
| SMTP | Mistral | 35 | 22 | 0 | 0.000 | 0.000 | 0.000 |

Table 36: TCP Partially Correct Transition Extraction Metrics

| Protocol | Model | TotalExtracted | TotalGT | Matched | Precision | Recall | F1-Score |
|---|---|---|---|---|---|---|---|
| TCP | DS-R1 | 15 | 20 | 11 | 0.733 | 0.550 | 0.629 |
| TCP | Gpt4o-Mini | 10 | 20 | 4 | 0.400 | 0.200 | 0.267 |
| TCP | Claude3 | 20 | 20 | 13 | 0.650 | 0.650 | 0.650 |
| TCP | Gemini2 | 45 | 20 | 5 | 0.111 | 0.250 | 0.154 |
| TCP | DS-V3 | 12 | 20 | 7 | 0.583 | 0.350 | 0.438 |
| TCP | QWQ | 9 | 20 | 5 | 0.556 | 0.250 | 0.345 |
| TCP | QWen3 | 9 | 20 | 5 | 0.556 | 0.250 | 0.345 |
| TCP | Gemma3 | 14 | 20 | 4 | 0.286 | 0.200 | 0.235 |
| TCP | Mistral | 11 | 20 | 2 | 0.182 | 0.100 | 0.129 |

Table 37: BGP Exact Transition Match Metrics

| Protocol | Model | TotalExtracted | TotalGT | Matched | Precision | Recall | F1-Score |
|---|---|---|---|---|---|---|---|
| BGP | DS-R1 | 20 | 26 | 14 | 0.700 | 0.538 | 0.609 |
| BGP | Gpt4o-Mini | 25 | 26 | 8 | 0.320 | 0.308 | 0.314 |
| BGP | Claude3 | 60 | 26 | 20 | 0.333 | 0.769 | 0.465 |
| BGP | Gemini2 | 101 | 26 | 25 | 0.248 | 0.962 | 0.394 |
| BGP | DS-V3 | 32 | 26 | 14 | 0.438 | 0.538 | 0.483 |
| BGP | QWQ | 7 | 26 | 2 | 0.286 | 0.077 | 0.121 |
| BGP | QWen3 | 11 | 26 | 5 | 0.455 | 0.192 | 0.270 |
| BGP | Gemma3 | 21 | 26 | 5 | 0.238 | 0.192 | 0.213 |
| BGP | Mistral | 9 | 26 | 0 | 0.000 | 0.000 | 0.000 |

Table 38: DCCP Exact Transition Match Metrics

| Protocol | Model | TotalExtracted | TotalGT | Matched | Precision | Recall | F1-Score |
|---|---|---|---|---|---|---|---|
| DCCP | DS-R1 | 29 | 25 | 11 | 0.379 | 0.440 | 0.407 |
| DCCP | Gpt4o-Mini | 55 | 25 | 4 | 0.073 | 0.160 | 0.100 |
| DCCP | Claude3 | 40 | 25 | 14 | 0.350 | 0.560 | 0.431 |
| DCCP | Gemini2 | 44 | 25 | 9 | 0.205 | 0.360 | 0.261 |
| DCCP | DS-V3 | 28 | 25 | 12 | 0.429 | 0.480 | 0.453 |
| DCCP | QWQ | 4 | 25 | 1 | 0.250 | 0.040 | 0.069 |
| DCCP | QWen3 | 15 | 25 | 5 | 0.333 | 0.200 | 0.250 |
| DCCP | Gemma3 | 18 | 25 | 4 | 0.222 | 0.160 | 0.186 |
| DCCP | Mistral | 9 | 25 | 0 | 0.000 | 0.000 | 0.000 |

Table 39: DHCP Exact Transition Match Metrics

| Protocol | Model | TotalExtracted | TotalGT | Matched | Precision | Recall | F1-Score |
|---|---|---|---|---|---|---|---|
| DHCP | DS-R1 | 20 | 19 | 13 | 0.650 | 0.684 | 0.667 |
| DHCP | Gpt4o-Mini | 26 | 19 | 9 | 0.346 | 0.474 | 0.400 |
| DHCP | Claude3 | 18 | 19 | 14 | 0.778 | 0.737 | 0.757 |
| DHCP | Gemini2 | 27 | 19 | 6 | 0.222 | 0.316 | 0.261 |
| DHCP | DS-V3 | 15 | 19 | 11 | 0.733 | 0.579 | 0.647 |
| DHCP | QWQ | 6 | 19 | 4 | 0.667 | 0.211 | 0.320 |
| DHCP | QWen3 | 4 | 19 | 0 | 0.000 | 0.000 | 0.000 |
| DHCP | Gemma3 | 10 | 19 | 0 | 0.000 | 0.000 | 0.000 |
| DHCP | Mistral | 5 | 19 | 2 | 0.400 | 0.105 | 0.167 |

Table 40: FTP Exact Transition Match Metrics

| Protocol | Model | TotalExtracted | TotalGT | Matched | Precision | Recall | F1-Score |
|---|---|---|---|---|---|---|---|
| FTP | DS-R1 | 23 | 24 | 5 | 0.217 | 0.208 | 0.213 |
| FTP | Gpt4o-Mini | 53 | 24 | 3 | 0.057 | 0.125 | 0.078 |
| FTP | Claude3 | 46 | 24 | 4 | 0.087 | 0.167 | 0.114 |
| FTP | Gemini2 | 51 | 24 | 6 | 0.118 | 0.250 | 0.160 |
| FTP | DS-V3 | 77 | 24 | 7 | 0.091 | 0.292 | 0.139 |
| FTP | QWQ | 7 | 24 | 3 | 0.429 | 0.125 | 0.194 |
| FTP | QWen3 | 14 | 24 | 1 | 0.071 | 0.042 | 0.053 |
| FTP | Gemma3 | 42 | 24 | 5 | 0.119 | 0.208 | 0.152 |
| FTP | Mistral | 33 | 24 | 3 | 0.091 | 0.125 | 0.105 |

Table 41: IMAP Exact Transition Match Metrics

| Protocol | Model | TotalExtracted | TotalGT | Matched | Precision | Recall | F1-Score |
|---|---|---|---|---|---|---|---|
| IMAP | DS-R1 | 35 | 11 | 4 | 0.114 | 0.364 | 0.174 |
| IMAP | Gpt4o-Mini | 56 | 11 | 3 | 0.054 | 0.273 | 0.090 |
| IMAP | Claude3 | 48 | 11 | 4 | 0.083 | 0.364 | 0.136 |
| IMAP | Gemini2 | 90 | 11 | 6 | 0.067 | 0.545 | 0.119 |
| IMAP | DS-V3 | 16 | 11 | 6 | 0.375 | 0.545 | 0.444 |
| IMAP | QWQ | 4 | 11 | 3 | 0.750 | 0.273 | 0.400 |
| IMAP | QWen3 | 4 | 11 | 2 | 0.500 | 0.182 | 0.267 |
| IMAP | Gemma3 | 17 | 11 | 3 | 0.176 | 0.273 | 0.214 |
| IMAP | Mistral | 20 | 11 | 5 | 0.250 | 0.455 | 0.323 |

Table 42: MQTT Exact Transition Match Metrics

| Protocol | Model | TotalExtracted | TotalGT | Matched | Precision | Recall | F1-Score |
|---|---|---|---|---|---|---|---|
| MQTT | DS-R1 | 16 | 17 | 1 | 0.062 | 0.059 | 0.061 |
| MQTT | Gpt4o-Mini | 37 | 17 | 2 | 0.054 | 0.118 | 0.074 |
| MQTT | Claude3 | 45 | 17 | 2 | 0.044 | 0.118 | 0.065 |
| MQTT | Gemini2 | 48 | 17 | 2 | 0.042 | 0.118 | 0.062 |
| MQTT | DS-V3 | 6 | 17 | 1 | 0.167 | 0.059 | 0.087 |
| MQTT | QWQ | 18 | 17 | 1 | 0.056 | 0.059 | 0.057 |
| MQTT | QWen3 | 6 | 17 | 1 | 0.167 | 0.059 | 0.087 |
| MQTT | Gemma3 | 19 | 17 | 1 | 0.053 | 0.059 | 0.056 |
| MQTT | Mistral | 37 | 17 | 2 | 0.054 | 0.118 | 0.074 |

Table 43: NNTP Exact Transition Match Metrics

| Protocol | Model | TotalExtracted | TotalGT | Matched | Precision | Recall | F1-Score |
|---|---|---|---|---|---|---|---|
| NNTP | DS-R1 | 73 | 16 | 1 | 0.014 | 0.062 | 0.022 |
| NNTP | Gpt4o-Mini | 68 | 16 | 3 | 0.044 | 0.188 | 0.071 |
| NNTP | Claude3 | 75 | 16 | 2 | 0.027 | 0.125 | 0.044 |
| NNTP | Gemini2 | 116 | 16 | 2 | 0.017 | 0.125 | 0.030 |
| NNTP | DS-V3 | 25 | 16 | 1 | 0.040 | 0.062 | 0.049 |
| NNTP | QWQ | 3 | 16 | 0 | 0.000 | 0.000 | 0.000 |
| NNTP | QWen3 | 6 | 16 | 0 | 0.000 | 0.000 | 0.000 |
| NNTP | Gemma3 | 14 | 16 | 0 | 0.000 | 0.000 | 0.000 |
| NNTP | Mistral | 4 | 16 | 0 | 0.000 | 0.000 | 0.000 |

Table 44: POP3 Exact Transition Match Metrics

| Protocol | Model | TotalExtracted | TotalGT | Matched | Precision | Recall | F1-Score |
|---|---|---|---|---|---|---|---|
| POP3 | DS-R1 | 21 | 18 | 8 | 0.381 | 0.444 | 0.410 |
| POP3 | Gpt4o-Mini | 33 | 18 | 10 | 0.303 | 0.556 | 0.392 |
| POP3 | Claude3 | 26 | 18 | 8 | 0.308 | 0.444 | 0.364 |
| POP3 | Gemini2 | 39 | 18 | 9 | 0.231 | 0.500 | 0.316 |
| POP3 | DS-V3 | 20 | 18 | 9 | 0.450 | 0.500 | 0.474 |
| POP3 | QWQ | 5 | 18 | 1 | 0.200 | 0.056 | 0.087 |
| POP3 | QWen3 | 8 | 18 | 4 | 0.500 | 0.222 | 0.308 |
| POP3 | Gemma3 | 38 | 18 | 6 | 0.158 | 0.333 | 0.214 |
| POP3 | Mistral | 29 | 18 | 9 | 0.310 | 0.500 | 0.383 |

Table 45: PPP Exact Transition Match Metrics

| Protocol | Model | TotalExtracted | TotalGT | Matched | Precision | Recall | F1-Score |
|---|---|---|---|---|---|---|---|
| PPP | DS-R1 | 111 | 27 | 10 | 0.090 | 0.370 | 0.145 |
| PPP | Gpt4o-Mini | 33 | 27 | 4 | 0.121 | 0.148 | 0.133 |
| PPP | Claude3 | 138 | 27 | 9 | 0.065 | 0.333 | 0.109 |
| PPP | Gemini2 | 126 | 27 | 19 | 0.151 | 0.704 | 0.248 |
| PPP | DS-V3 | 21 | 27 | 3 | 0.143 | 0.111 | 0.125 |
| PPP | QWQ | 5 | 27 | 0 | 0.000 | 0.000 | 0.000 |
| PPP | QWen3 | 21 | 27 | 1 | 0.048 | 0.037 | 0.042 |
| PPP | Gemma3 | 9 | 27 | 0 | 0.000 | 0.000 | 0.000 |
| PPP | Mistral | 12 | 27 | 0 | 0.000 | 0.000 | 0.000 |

Table 46: PPTP Exact Transition Match Metrics

| Protocol | Model | TotalExtracted | TotalGT | Matched | Precision | Recall | F1-Score |
|---|---|---|---|---|---|---|---|
| PPTP | DS-R1 | 18 | 19 | 12 | 0.667 | 0.632 | 0.649 |
| PPTP | Gpt4o-Mini | 29 | 19 | 2 | 0.069 | 0.105 | 0.083 |
| PPTP | Claude3 | 56 | 19 | 3 | 0.054 | 0.158 | 0.080 |
| PPTP | Gemini2 | 23 | 19 | 0 | 0.000 | 0.000 | 0.000 |
| PPTP | DS-V3 | 34 | 19 | 12 | 0.353 | 0.632 | 0.453 |
| PPTP | QWQ | 2 | 19 | 0 | 0.000 | 0.000 | 0.000 |
| PPTP | QWen3 | 6 | 19 | 3 | 0.500 | 0.158 | 0.240 |
| PPTP | Gemma3 | 6 | 19 | 0 | 0.000 | 0.000 | 0.000 |
| PPTP | Mistral | 12 | 19 | 1 | 0.083 | 0.053 | 0.065 |

Table 47: RTSP Exact Transition Match Metrics

| Protocol | Model | TotalExtracted | TotalGT | Matched | Precision | Recall | F1-Score |
|---|---|---|---|---|---|---|---|
| RTSP | DS-R1 | 41 | 33 | 29 | 0.707 | 0.879 | 0.784 |
| RTSP | Gpt4o-Mini | 70 | 33 | 2 | 0.029 | 0.061 | 0.039 |
| RTSP | Claude3 | 89 | 33 | 20 | 0.225 | 0.606 | 0.328 |
| RTSP | Gemini2 | 78 | 33 | 13 | 0.167 | 0.394 | 0.234 |
| RTSP | DS-V3 | 39 | 33 | 32 | 0.821 | 0.970 | 0.889 |
| RTSP | QWQ | 2 | 33 | 0 | 0.000 | 0.000 | 0.000 |
| RTSP | QWen3 | 10 | 33 | 8 | 0.800 | 0.242 | 0.372 |
| RTSP | Gemma3 | 24 | 33 | 5 | 0.208 | 0.152 | 0.175 |
| RTSP | Mistral | 33 | 33 | 0 | 0.000 | 0.000 | 0.000 |

Table 48: SIP Exact Transition Match Metrics

| Protocol | Model | TotalExtracted | TotalGT | Matched | Precision | Recall | F1-Score |
|---|---|---|---|---|---|---|---|
| SIP | DS-R1 | 58 | 20 | 12 | 0.207 | 0.600 | 0.308 |
| SIP | Gpt4o-Mini | 127 | 20 | 9 | 0.071 | 0.450 | 0.122 |
| SIP | Claude3 | 164 | 20 | 0 | 0.000 | 0.000 | 0.000 |
| SIP | Gemini2 | 180 | 20 | 18 | 0.100 | 0.900 | 0.180 |
| SIP | DS-V3 | 43 | 20 | 13 | 0.302 | 0.650 | 0.413 |
| SIP | QWQ | 5 | 20 | 0 | 0.000 | 0.000 | 0.000 |
| SIP | QWen3 | 11 | 20 | 3 | 0.273 | 0.150 | 0.194 |
| SIP | Gemma3 | 47 | 20 | 2 | 0.043 | 0.100 | 0.060 |
| SIP | Mistral | 30 | 20 | 0 | 0.000 | 0.000 | 0.000 |

Table 49: SMTP Exact Transition Match Metrics

| Protocol | Model | TotalExtracted | TotalGT | Matched | Precision | Recall | F1-Score |
|---|---|---|---|---|---|---|---|
| SMTP | DS-R1 | 45 | 22 | 0 | 0.000 | 0.000 | 0.000 |
| SMTP | Gpt4o-Mini | 52 | 22 | 0 | 0.000 | 0.000 | 0.000 |
| SMTP | Claude3 | 40 | 22 | 0 | 0.000 | 0.000 | 0.000 |
| SMTP | Gemini2 | 32 | 22 | 1 | 0.031 | 0.045 | 0.037 |
| SMTP | DS-V3 | 18 | 22 | 1 | 0.056 | 0.045 | 0.050 |
| SMTP | QWQ | 11 | 22 | 0 | 0.000 | 0.000 | 0.000 |
| SMTP | QWen3 | 7 | 22 | 0 | 0.000 | 0.000 | 0.000 |
| SMTP | Gemma3 | 6 | 22 | 0 | 0.000 | 0.000 | 0.000 |
| SMTP | Mistral | 35 | 22 | 0 | 0.000 | 0.000 | 0.000 |

Table 50: TCP Exact Transition Match Metrics

| Protocol | Model | TotalExtracted | TotalGT | Matched | Precision | Recall | F1-Score |
|---|---|---|---|---|---|---|---|
| TCP | DS-R1 | 15 | 20 | 9 | 0.600 | 0.450 | 0.514 |
| TCP | Gpt4o-Mini | 10 | 20 | 4 | 0.400 | 0.200 | 0.267 |
| TCP | Claude3 | 20 | 20 | 12 | 0.600 | 0.600 | 0.600 |
| TCP | Gemini2 | 45 | 20 | 1 | 0.022 | 0.050 | 0.031 |
| TCP | DS-V3 | 12 | 20 | 7 | 0.583 | 0.350 | 0.438 |
| TCP | QWQ | 9 | 20 | 3 | 0.333 | 0.150 | 0.207 |
| TCP | QWen3 | 9 | 20 | 4 | 0.444 | 0.200 | 0.276 |
| TCP | Gemma3 | 14 | 20 | 3 | 0.214 | 0.150 | 0.176 |
| TCP | Mistral | 11 | 20 | 2 | 0.182 | 0.100 | 0.129 |

