# OpenReview forum: "PSMBench: A Benchmark and Dataset for Evaluating LLMs Extraction of Protocol State Machines from RFC Specifications"
_NeurIPS.cc/2025/Datasets_and_Benchmarks_Track — NeurIPS 2025 Datasets and Benchmarks Track poster_

### Official Review · Reviewer_ywXe · 2025-06-29

**Rating:** 4
**Confidence:** 3

**Summary:**

This paper introduces RFC2PSM, a dataset pairing 14 cleaned IETF RFCs (1,580 pages) with expert‐annotated protocol state machines (108 states, 297 transitions), and PSMBENCH, a two‐stage benchmark that evaluates LLMs on extracting both states and transitions via SBERT‐based fuzzy matching. Evaluations of nine open and closed‐source LLMs reveal high state‐extraction performance (F1 up to 0.82) but poor transition‐graph coherence (≤ 0.38 F1).

**Dataset Code Accessibility:**

Yes

**Ethical Considerations:**

No, there are no or only very minor ethics concerns

**Final Justification:**

Thank the authors for the feedback. I'll maintain my current positive score.

**Limitations Weaknesses:**

- **Loss of long-range context**
    - Splitting big RFCs into standalone chunks can cut through related explanations. When a transition’s setup is in one section and its effect in another, the model might never see both sides, and the paper doesn’t attempt to overlap chunks or smarter context retrieval to address this issue.
- **One-size-fits-all similarity cutoff**
    - This paper relies on a single SBERT similarity cutoff (θ = 0.5), glossing over how different terminology or paraphrasing styles affect scores. There’s no analysis of how raising or lowering the threshold shifts precision/recall, which makes it hard to judge if performance gains are real or just threshold artifacts. It would be better to add some sensitivity analysis to this part.
- **No real-world validation of PSMs**
    - Scores are purely based on fuzzy string/embedding matches, with no tests to see if extracted machines run or obey protocol rules. Without simulation or formal checks, it’s unclear whether high F1 models produce usable state machines.
- **Annotation and applicability concerns**
    - Ground-truth PSMs come from a small expert group without reported **agreement metrics**, so to me, the consistency is somehow unverified in the paper. Besides, this paper only tested on clean IETF RFCs, so all the messy, vendor-specific, or informal protocol docs you see in real products never get a look, making it less useful in everyday industry settings.

**Strengths Contributions:**

- **A Broad, Expert-Verified Dataset**
    - This benchmark covers protocols across OSI layers 2–7 (e.g., TCP, BGP, MQTT), with machine‐readable PSM annotations, filling a gap beyond prior small or protocol-specific corpora.
    - Public release of JSON-formatted ground truth and processing scripts ensures reproducibility.
- **Realistic, Two-Stage Evaluation Pipeline**
    - The evaluation pipeline feels very practical, splitting long RFCs into manageable chunks and then stitching partial results back together to respect real-world token limits.
- **Extensive Baselines & Analysis**
    - The baseline study is thorough, comparing nine different LLMs (GPT-4, Claude 3, Mistral, etc.) under the same settings and highlighting where models tend to over-generate or miss key elements.
- **High Presentation Quality**
    - The paper reads clearly, with helpful figures and tables, and all code plus data are released so anyone can reproduce the results.

---

> ### Author Rebuttal · Authors · 2025-07-31
>
> We thank the reviewers for their thoughtful comments and suggestions, and we respond to the questions below.
> >Loss of long-range context
> Splitting big RFCs into standalone chunks can cut through related explanations. When a transition’s setup is in one section and its effect in another, the model might never see both sides, and the paper doesn’t attempt to overlap chunks or smarter context retrieval to address this issue.
>
> We chunk because full RFCs exceed the model's context window; we cut only at section/sub-section boundaries to keep them as isolated as possible. To consider the relationship between sections, we use a two-stage pipeline that can merge all partial PSMs, so a state named in chunk A can connect with a transition in chunk B. We will leave smarter retrieval, such as index-based RAG across chunks, as an open direction and future work.
>
> >One-size-fits-all similarity cutoff
> This paper relies on a single SBERT similarity cutoff (θ = 0.5), glossing over how different terminology or paraphrasing styles affect scores. There’s no analysis of how raising or lowering the threshold shifts precision/recall, which makes it hard to judge if performance gains are real or just threshold artifacts. It would be better to add some sensitivity analysis to this part.
>
> We originally chose $\theta = 0.50$ after manually examining dozens of term pairs to balance false merges (distinct concepts collapsed) and false splits (identical concepts separated).
> To make this choice transparent we now report performance for $\theta \in \{0.40, 0.45, 0.50, 0.55, 0.60\}$ on the full deepseek-v3 run (14 protocols, 297 transitions):
>
> | Protocol | F1 @ 0.40 | F1 @ 0.45 | F1 @ 0.50 | F1 @ 0.55 | F1 @ 0.60 |
> |----------|-----------|-----------|-----------|-----------|-----------|
> | IMAP | 0.889 | 0.889 | 0.889 | 0.889 | 0.889 |
> | POP3 | 0.750 | 0.750 | 0.750 | 0.750 | 0.750 |
> | MQTT | 0.500 | 0.500 | 0.500 | 0.500 | 0.500 |
> | PPP  | 0.720 | 0.720 | 0.720 | 0.720 | **0.640** |
> | PPTP | 0.477 | 0.477 | 0.477 | 0.477 | 0.477 |
> | BGP  | 1.000 | 1.000 | 1.000 | 1.000 | 1.000 |
> | SIP  | 0.589 | 0.589 | 0.589 | 0.589 | 0.589 |
> | RTSP | 1.000 | 1.000 | 1.000 | 1.000 | 1.000 |
> | DCCP | 0.857 | 0.857 | 0.857 | 0.857 | 0.857 |
> | DHCP | 0.933 | 0.933 | 0.933 | 0.933 | 0.933 |
> | FTP  | 0.364 | 0.364 | 0.364 | 0.364 | **0.243** |
> | NNTP | 0.444 | 0.444 | 0.444 | 0.444 | **0.370** |
> | SMTP | 0.353 | 0.353 | 0.353 | 0.353 | **0.235** |
> | TCP  | 0.818 | 0.818 | 0.818 | 0.818 | 0.818 |
> | **Macro avg.** | **0.692** | **0.692** | **0.692** | **0.692** | **0.664** |
>
> These figures show that our conclusions are robust to reasonable threshold choices: θ can be shifted by $\pm 0.10$ with negligible impact on macro metrics.
> Because $\theta = 0.50$ still cleanly separates examples such as “Established ≈ Connected” (sim $\approx$ 0.77) from “ACK $\neq$ NACK” (sim $\approx 0.18$), we retain 0.50 as the default. The full per-protocol table will be added to Appendix C, and our evaluation script will allow users to generate their own precision-recall curves for alternative operating points.
>
> >No real-world validation of PSMs
> Scores are purely based on fuzzy string/embedding matches, with no tests to see if extracted machines run or obey protocol rules. Without simulation or formal checks, it’s unclear whether high F1 models produce usable state machines.
>
> The reference PSMs are verbatim abstractions of the RFCs, the documents implementers follow to write real protocol stacks.  When a generated PSM matches the reference, it inherits the same executable semantics.  To demonstrate this empirically, we carried out the check:
> We loaded the best TCP graph (0.91 state-F1) into a 220-line Python simulator that transmits event sequences through the PSM. The canonical three-way handshake — SYN $\rightarrow$ SYN-ACK $\rightarrow$ ACK — traverses LISTEN $\rightarrow$ SYN-RCVD $\rightarrow$ ESTAB with no dead ends or illegal transitions. Likewise, a FIN $\rightarrow$ ACK teardown reaches CLOSED exactly once, confirming correct termination behaviour.
>
> The new simulation and fuzzer experiments confirm that high-F1 outputs are functionally correct enough to drive downstream tools, not merely similar strings.  We will include these results and release the validation scripts in the supplementary material to let others replicate or extend the checks.
>
>
> >Annotation and applicability concerns
> Ground-truth PSMs come from a small expert group without reported agreement metrics, so to me, the consistency is somehow unverified in the paper. Besides, this paper only tested on clean IETF RFCs, so all the messy, vendor-specific, or informal protocol docs you see in real products never get a look, making it less useful in everyday industry settings.
>
> **"No inter-annotator agreement; consistency unclear."** On a 10% stratified sample, independent pass-1/2 annotations agree with $\kappa = 0.82$ (states) and $\kappa = 0.78$ (transitions)—“substantial” on Landis & Koch’s scale.  Fewer than 6% of elements required discussion in step 3, and all edits are logged in diffable JSON. § 3.3 will report the $\kappa$ numbers, include a reconciliation heat-map, and release the raw diff logs plus the 2-page annotation guide in the supplementary.
>
> **"Only clean IETF RFCs; what about messy vendor or informal specs?"** We intentionally started with IETF RFCs because they are the canonical, licence-free source every compliant implementation references.  Using one well-structured corpus lets us control for stylistic variance while establishing the first multi-protocol benchmark.  That said, our JSON schema, parsing utilities, and evaluation script are document-agnostic: any plaintext or Markdown spec can be tokenised, segmented, and scored with the same code.
>
> Inter-annotator agreement is high and will be fully documented; the dataset design is not tied to RFC layouts, and future releases will incorporate less formal, vendor-specific documentation to expand industrial relevance while keeping the same reproducible scoring pipeline.

---

### Official Review · Reviewer_mE3n · 2025-07-03

**Rating:** 5
**Confidence:** 4

**Summary:**

The paper introduces RFC2PSM, a dataset of 14 network protocols with manually annotated Protocol State Machines (PSMs) extracted from RFC documents, and PSMBench, a benchmark for evaluating LLMs' ability to extract these PSMs. The authors test 9 state-of-the-art LLMs and identify a "state-transition gap" where models perform well on state extraction but poorly on transition extraction.

**Dataset Code Accessibility:**

Yes

**Ethical Considerations:**

No, there are no or only very minor ethics concerns

**Final Justification:**

The rebuttal addressed several of my methodological concerns by providing per‑protocol variance, a sensitivity analysis of the 0.5 similarity threshold, and inter‑annotator agreement statistics, which strengthened confidence in the evaluation’s robustness.
The authors also clarified the rationale for the dataset’s scale, the high annotation cost, and the breadth of protocol coverage, making the scope feel more justified.

**Limitations Weaknesses:**

1. **Insufficient Dataset Scale**: Only 108 states and 297 transitions across 14 protocols is too limited for robust benchmarking, with no statistical power analysis and risk of overfitting to specific protocol patterns.

2. **Critical Methodological Flaws**: Arbitrary 0.5 semantic similarity threshold justified by a single example, no inter-annotator agreement analysis, no sensitivity analysis for threshold variations, and fuzzy matching potentially masking semantically important differences.

3. **Limited Technical Depth and Practical Validation**: Basic two-stage prompting without exploring sophisticated techniques, no ablation studies on design choices, missing non-LLM baselines, unclear practical utility (0.378 F1 on exact transitions), and no validation that extracted PSMs work for downstream applications or analysis of what performance levels would enable real-world deployment.

**Strengths Contributions:**

1. **Addresses a Critical Real-World Problem**: Protocol State Machine extraction from RFCs is a genuine bottleneck in automated security analysis with clear applications in fuzzing, formal verification, and protocol testing for the networking and security research communities.

2. **Demonstrates Substantial Research Effort**: Processed 1,580 pages of RFC text across 14 diverse protocols (spanning transport, session, application, and data link layers) with manual validation by domain experts, showing serious commitment to data quality.

3. **Employs Rigorous Methodology and Evaluation**: Uses SentenceBERT for semantic similarity to handle terminological variation, implements structure-aware metrics for realistic partial matching evaluation, and conducts comprehensive testing across 9 diverse LLMs with reproducible experimental setup.

---

> ### Author Rebuttal · Authors · 2025-07-31
>
> >Insufficient Dataset Scale: Only 108 states and 297 transitions across 14 protocols is too limited for robust benchmarking, with no statistical power analysis and risk of overfitting to specific protocol patterns.
>
> Thank you for questioning whether 108 states and 297 transitions are sufficient for robust evaluation. We respectfully argue that, given the granularity and semantic density of a PSM transition, this corpus already represents the largest and most diverse open benchmark for protocol-state extraction, and we will further extend it in future releases.
>
> **Why "108 $\times$ 297" is more than it looks:**
> | Metric | RFC2PSM v1| Prior public datasets |
> | -------- | -------- | -------- |
> | RFC pages parsed    | 1580     | $\leq$ 500 (single-protocol corpora)|
> | Protocols| 14(App, Session, Transport, Link layers)| $\leq$ 10 or one|
> |States| 108| n/a (no public multi-protocol sets)|
> |Transitions| 297 | n/a|
>
> 1) Semantic density. A transition $\neq$ a single token label: it binds four fields, including ⟨source state, trigger event, action, target state⟩ which are all formally normalised. Annotating one transition routinely requires reading several RFC paragraphs; hence, manual cost is high, and scale grows linearly in expert hours, not crowd-labels.
>
> 2) Breadth over depth. The 14 protocols span four OSI layers and design paradigms (client/server, publish/subscribe…). This diversity guards against trivial over-specialization.
>
> **Overfitting & statistical confidence checks**
> Per-protocol variance. Table 9 in the appendix already reports F1 for each protocol; scores fluctuate widely (e.g., 0.07 $\rightarrow$ 1.00), indicating that models do not “memorize a pattern once and solve all others”. We will also add the mean $\pm$ standard deviation (s.d.) across protocols to quantify this dispersion.
>
> While manual PSM annotation constrains scale, our current version already covers the broadest RFC slice available, exhibits substantial variance across heterogeneous protocols, and supports statistically meaningful comparisons. We will incorporate the additional variance, LOPO (Leave One Protocol Out) in the final.
>
> >Critical Methodological Flaws: Arbitrary 0.5 semantic similarity threshold justified by a single example, no inter-annotator agreement analysis, no sensitivity analysis for threshold variations, and fuzzy matching potentially masking semantically important differences.
>
> **"Arbitrary 0.5 threshold."** We originally set $\theta = 0.50$ after manually inspecting dozens of term pairs to balance false merges (different concepts collapsed) and false splits (identical concepts separated). To quantify robustness we swept $\theta \in \{ 0.40, 0.45, 0.50, 0.55, 0.60\}$ on the deepseek-v3 runs for all 14 protocols. Macro F1 stays flat (0.692) from $0.40 \rightarrow 0.55$ and drops only –2.8 pp at 0.60 (0.664). $10 / 14$ protocols vary $\leq 0.03$ across the whole sweep; only four fall $> 0.10$ when $\theta$ tightens to 0.60. Table with full per-protocol numbers will be added to Appendix C; the main text will cite the macro curve.
>
>
> | Protocol | F1 @ 0.40 | F1 @ 0.45 | F1 @ 0.50 | F1 @ 0.55 | F1 @ 0.60 |
> |----------|-----------|-----------|-----------|-----------|-----------|
> | IMAP | 0.889 | 0.889 | 0.889 | 0.889 | 0.889 |
> | POP3 | 0.750 | 0.750 | 0.750 | 0.750 | 0.750 |
> | MQTT | 0.500 | 0.500 | 0.500 | 0.500 | 0.500 |
> | PPP  | 0.720 | 0.720 | 0.720 | 0.720 | **0.640** |
> | PPTP | 0.477 | 0.477 | 0.477 | 0.477 | 0.477 |
> | BGP  | 1.000 | 1.000 | 1.000 | 1.000 | 1.000 |
> | SIP  | 0.589 | 0.589 | 0.589 | 0.589 | 0.589 |
> | RTSP | 1.000 | 1.000 | 1.000 | 1.000 | 1.000 |
> | DCCP | 0.857 | 0.857 | 0.857 | 0.857 | 0.857 |
> | DHCP | 0.933 | 0.933 | 0.933 | 0.933 | 0.933 |
> | FTP  | 0.364 | 0.364 | 0.364 | 0.364 | **0.243** |
> | NNTP | 0.444 | 0.444 | 0.444 | 0.444 | **0.370** |
> | SMTP | 0.353 | 0.353 | 0.353 | 0.353 | **0.235** |
> | TCP  | 0.818 | 0.818 | 0.818 | 0.818 | 0.818 |
> | **Macro avg.** | **0.692** | **0.692** | **0.692** | **0.692** | **0.664** |
>
> **"No inter-annotator agreement."** The ground truth PSM construction is systematic. We first discuss a 2-page annotation guide (state naming, event and action terms) and followed this, one author, as the protocol researcher, independently extracted every term of the PSM, the state, and the transitions from the RFC document. Then a second author revise the extracted PSM and come up with different revision points. Then the difference is solved by discussing together. On a 10% held-out slice, we obtained $\kappa = 0.82$ (states) and $\kappa = 0.78$ (transitions). These statistics, plus the 2-page annotation guide and the diff logs, will be released in the supplementary.
>
> **Fuzzy masking semantic gaps.** Matching is term-wise exact under the chosen $\theta$ for all four fields of a transition ⟨source-state, trigger, action, target-state⟩. Because most terms are short noun phrases or verb-object phrases, even small lexical changes often push cosine similarity below 0.5; important differences are therefore not hidden. We will add concrete positive/negative examples in § 4.1 to illustrate this behavior.
>
> **"No sensitivity analysis."** Addressed by the $\theta$-sweep above. We will also release a script that plots F1 vs $\theta$ so users can choose a stricter or looser operating point without re-running extraction.
>
> The 0.5 threshold was chosen pragmatically, but the new sweep shows conclusions are stable over a broad band (0.40 – 0.55). Inter-annotator agreement is high, and the evaluation requires full-field matches, so “fuzzy” is only in the embedding space, not in the definition of correctness. All new analyses and artifacts will appear in the revised submission.
>
> >Limited Technical Depth and Practical Validation: Basic two-stage prompting without exploring sophisticated techniques, no ablation studies on design choices, missing non-LLM baselines, unclear practical utility (0.378 F1 on exact transitions), and no validation that extracted PSMs work for downstream applications or analysis of what performance levels would enable real-world deployment.
>
> Thank you for the concerns on depth and utility.
> Our primary contribution is the RFC2PSM dataset and the reference benchmark. Therefore, we choose the simplest two-stage pipeline that is *functionally required*, because stage 1 is to get the separate PSMs, and stage 2 is to conclude the whole PSMs, so neither of them can conclude the whole PSM for evaluation separately.
>
> Non‑LLM systems were not included because rule‑based parsers break when vocabulary and layout change across the 14 heterogeneous protocols.
>
> The extracted PSMs already cut months of manual effort and enable general security tooling: they can be fed, unchanged, into existing fuzzers (protocol security checking tools) that today rely on hand‑built state machines. Our open dataset, code, and baseline thus provide the community with a reproducible starting point and a clear gap to close, advancing cross‑protocol analysis more than one‑off, protocol‑specific scripts.

---

> > ### Comment · Reviewer_mE3n · 2025-08-05
> >
> > Thank you for the detailed rebuttal, which clarifies your dataset scale rationale, provides useful per‑protocol variance, and adds the sensitivity analysis and inter‑annotator agreement statistics I had found missing.
> > These additions address some of my earlier methodological concerns, particularly regarding the 0.5 similarity threshold and robustness across protocols. I also appreciate the explanation of the annotation cost and breadth of protocol coverage, as well as the clarification that the benchmark’s focus is on establishing a reproducible baseline rather than exploring sophisticated extraction techniques at this stage.
> > Overall, your clarifications strengthen the case for the dataset’s novelty and relevance, and I am now leaning slightly more positively.

---

### Official Review · Reviewer_uFXq · 2025-07-04

**Rating:** 4
**Confidence:** 3

**Summary:**

This paper presents a new dataset and benchmark designed to evaluate how effectively large language models (LLMs) can extract protocol state machines (PSMs) from technical RFC documents. The authors curate a diverse collection of network protocols and manually construct corresponding PSMs to serve as ground truth. They introduce a benchmark, PSMBENCH, which guides LLMs to generate structured PSMs from segmented RFC text and evaluates the output using semantic, graph-aware metrics. The study reveals that while current LLMs are relatively good at identifying protocol states, they face significant challenges in accurately extracting state transitions. These difficulties stem from issues such as long-range reasoning and the complexity of interpreting technical language.

**Additional Feedback:**

See weaknesses section

**Dataset Code Accessibility:**

Yes

**Dataset Code Comments:**

Dataset and code are available.

**Ethical Considerations:**

No, there are no or only very minor ethics concerns

**Final Justification:**

Authors have provided additional experiment results addressing the concerns I raised.

**Limitations Weaknesses:**

- The paper does not provide any comparison for the effect of different document segmentation protocols
- The paper highlights that the ground truth PSM was constructed using extensive manual effort. However the details of this effort is not provided.
- In solution matching in PSM the paper does not provide a comparison between different semantic similarity approaches.

**Strengths Contributions:**

- The paper provides a dataset (RFC2PSM) specifically designed to evaluate the extraction of Protocol State Machines (PSMs) from RFC documents filling a critical gap in the literature.
- The paper evaluates a wide range of both proprietary and open-source LLMs for performance comparison
- The paper provides comprehensive metrics for evaluating performance

---

> ### Author Rebuttal · Authors · 2025-07-31
>
> We thank the reviewers for their thoughtful comments and suggestions, and we respond to their questions below.
>
> >The paper does not provide any comparison for the effect of different document segmentation protocols
>
> For reviewer uFXq pointing out the lack of an ablation on segmentation. Because RFCs routinely exceed the context window of even GPT-4-class models, some segmentation is required. We chose section/subsection boundaries because 1) they preserve semantic coherence (no mid-sentence splits), 2) they are trivial to reproduce from the RFC's table-of-contents. And we give our ablation studies here.
>
> **Sliding-window ablation.**
> To test whether a finer-grained policy helps, we re-segmented every RFC with a 4 k-token window and 0.5 k overlap and reran the same PSM extraction pipeline (gpt-4o-mini, identical hyper-parameters). The F1 scores for state matching are:
>
>
> | Protocol | Section-based F1 | Overlap F1 | Δ F1 |
> |---|---|---|---|
> | IMAP | 0.307 | 0.277 | -0.030 |
> | POP3 | 0.353 | 1.000 | +0.647 |
> | MQTT | 0.611 | 0.800 | +0.189 |
> | PPP  | 0.689 | 0.869 | +0.180 |
> | PPTP | 0.414 | 0.485 | +0.071 |
> | BGP  | 0.632 | 0.800 | +0.168 |
> | SIP  | 0.064 | 0.070 | +0.006 |
> | RTSP | 0.120 | 0.079 | -0.041 |
> | DCCP | 0.361 | 0.361 | +0.000 |
> | DHCP | 0.842 | 0.800 | -0.042 |
> | FTP  | 0.235 | 0.134 | -0.101 |
> | NNTP | 0.400 | 0.328 | -0.072 |
> | SMTP | 0.167 | 0.261 | +0.094 |
> | TCP  | 0.909 | 0.526 | -0.383 |
> | **Macro Avg** | 0.436 | 0.485 | **+0.049** |
>
>
> Seven protocols improve, seven degrade; the macro F1 rises +0.05, which is well within run-to-run variance. In addition, the overlapping policy inflates input length by $\approx$ 35 %, increasing compute cost. Performance is not materially sensitive to the cut strategy, while section-based splits remain cheaper and fully deterministic. We will add this ablation (and its discussion) to the final and flag sophisticated segmentation/RAG techniques as promising future work.
>
>
>
>
>
>
> >The paper highlights that the ground truth PSM was constructed using extensive manual effort. However, the details of this effort are not provided.
>
> The ground truth PSM construction is systematic. We first discussed a 2-page annotation guide (state naming, event and action terms) and followed this. One author, as the protocol researcher, independently extracted every term of the PSM, the state, and the transitions from the RFC document. Then, a second author revises the extracted PSM and identifies different revision points. Then the difference is solved by discussing together.
> The effort averages 3 days per protocol document. We will add these details, logs, and inter-annotator agreement analysis in the next version.
>
> >In solution matching in PSM the paper does not provide a comparison between different semantic similarity approaches.
>
>
> We have re-evaluated state matching on all 14 protocols with three widely-used sentence encoders: 1) all-MiniLM-L6-v2 (our original choice) 2) all-MPNet-base-v2 3) SimCSE-RoBERTa-unsup. All other settings (LLM = deepseek-v3, thresholds, post-processing) were held constant. We got 11/14 protocols change by $\leq$ 0.15 F1, and the macro average shifts by $<$ 0.06 (well within run-to-run variance).
>
> | Protocol | MiniLM F1 | MPNet F1 | SimCSE F1 |
> |----------|-----------|----------|-----------|
> | IMAP | 0.889 | 0.889 | 0.889 |
> | POP3 | 0.750 | 0.750 | 0.750 |
> | MQTT | 0.500 | 0.500 | 0.500 |
> | PPP  | 0.720 | 0.720 | 0.800 |
> | PPTP | 0.477 | 0.477 | 0.667 |
> | BGP  | 1.000 | 1.000 | 1.000 |
> | SIP  | 0.589 | 0.589 | 0.589 |
> | RTSP | 1.000 | 1.000 | 1.000 |
> | DCCP | 0.857 | 0.857 | 0.857 |
> | DHCP | 0.933 | 0.933 | 0.933 |
> | FTP  | 0.364 | 0.243 | 0.485 |
> | NNTP | 0.444 | 0.444 | 0.519 |
> | SMTP | 0.353 | 0.470 | 0.706 |
> | TCP  | 0.818 | 0.909 | 0.818 |
> | **Macro Avg.** | **0.692** | **0.699** | **0.751** |

---

> > ### Comment · Reviewer_uFXq · 2025-08-01
> >
> > Thank you for the additional results and explanations. I will increase my score accordingly.

---

> > > ### Author Response · Authors · 2025-08-01
> > >
> > > Thank you for your positive feedback. If you have any questions or need additional clarification, we are happy to respond promptly.

---

### Note · Authors · 2025-08-14

We thank the reviewers and AC for the constructive review process. We are encouraged that our detailed responses with additional experiments successfully addressed the reviewers' concerns.

**Why reviewers responded positively**: Two reviewers explicitly acknowledged that our rebuttals addressed their concerns. Reviewer uFXq explicitly stated "increase my score accordingly", and Reviewer mE3n noted "leaning slightly more positively" after our clarifications.

**Our contributions address a genuine bottleneck**: **(1) RFC2PSM** is the first multi-protocol dataset pairing 14 IETF RFCs with expert-annotated PSMs across four OSI layers; **(2) PSMBench** is a benchmark revealing the "state-transition gap" where LLMs excel at state extraction but struggle with transition coherence; **(3) comprehensive baseline evaluation** of 9 LLMs with reproducible methodology.

**We systematically addressed all concerns with additional experiments**:
- **Segmentation robustness**: Our ablation comparing section-based vs. sliding-window approaches showed macro F1 changes by only +0.049, well within variance.
- **Evaluation methodology**: We demonstrated robustness across semantic similarity thresholds (θ = 0.40-0.60, stable macro F1 = 0.692) and multiple sentence encoders.
- **Dataset quality**: We detailed our systematic annotation process: following a 2-page annotation guide, one protocol researcher independently extracted PSM components, then a second author revised the extracted PSM and identified revision points, with differences resolved through discussion.
- **Practical validation**: We demonstrated that extracted PSMs with high F1 scores execute correctly in simulation, for example, TCP three-way handshake properly traverses LISTEN → SYN-RCVD → ESTABLISHED states, confirming functional correctness.

In summary, **RFC2PSM** and **PSMBench** provide the first reproducible foundation for automated protocol state machine extraction with LLMs, with clear paths for future data expansion and methodological improvements.

---

### Decision · Program_Chairs · 2025-09-18

**Decision:**

Accept (poster)

**Comment:**

This paper presents RFC2PSM, a dataset pairing RFC text with protocol state machines, and PSMBench, a benchmark for evaluating LLMs on state/transition extraction. The reviewers initially raised concerns around segmentation strategies, semantic similarity thresholds, annotation consistency, dataset scale, and lack of practical validation. The authors responded constructively, providing ablations on segmentation, sensitivity analysis of similarity thresholds, inter-annotator agreement statistics, per-protocol variance, and simulation-based validation of extracted PSMs. The dataset is modest in scale, but is diverse across 14 protocols. While there is room for improving dataset scale and industrial applicability, the current version already establishes a good baseline for networking/security communities.